

# Cyanobacteria Blooms in the Baltic Sea: A Review of Models and Facts

Britta Munkes[1], Ulrike Löptien[1,2], and Heiner Dietze[1,2]

[1]GEOMAR, Helmholtz Centre for Ocean Research Kiel, Düsternbrooker Weg 20, D-24105 Kiel, Germany.
[2]Institute of Geosciences, Christian-Albrechts-University of Kiel, Ludewig-Meyn-Str. 10, 24 118 Kiel, Germany

**Correspondence:** Britta Munkes (bmunkes@geomar.de)

**Abstract.** The ecosystem of the Baltic Sea is endangered by eutrophication. This has triggered expensive international management efforts. Some of these efforts are impeded by natural processes such as nitrogen-fixing cyanobacteria blooms that add bioavailable nitrogen to the already over-fertilised system and thereby enhance primary production, export of organic matter to depth and associated oxygen consumption. Controls of cyanobacteria blooms are not comprehensively understood and this

adds to the uncertainty of model-based projections into the warming future of the Baltic Sea. Here we review our current understanding of cyanobacteria bloom dynamics. We summarise published field studies, laboratory experiments and dissect the basic principles ingrained in state-of-the-art coupled ocean-circulation biogeochemical models.

## 1  Introduction

The Baltic Sea is a shallow, brackish and semi-enclosed sea in central Northern Europe. It's drainage basin is densely populated by around 84 million people. Their footprint exerts pressure on the ecosystem (Unger et al., 2013; Hannerz and Destouni, 2006). One, particularly severe, problem is eutrophication. Antropogenic nutrients enter the Baltic Sea via rivers and air-sea fluxes (Helcom, 2018, 2014). Starting with the first Helsinki Convention in 1974, several international environmental management plans have been put to work; so far with varying degrees of success (e.g. Helsinki Convention, EU Marine Strategy Framework

Directive, Baltic Sea Action Plan). On the one hand the Baltic Sea is one of the best investigated and managed seas in the world (Helcom, 2018b) where international efforts have been successful in reducing nutrient loads considerably (Helcom, 2018). On the other hand, despite all the resource-intensive management efforts, the state of the ecosystem has not significantly improved yet (HOLAS II core indicator report, 2017; Gustafsson et al. (2017)). One major cause is considered to be related to sedimentary processes. Another concern are nitrogen fixing cyanobacteria blooms, which are suspected to become more

prevalent with warming temperatures.

The ability of cyanobacteria to utilise dinitrogen, a virtually unlimited resource in air, and to convert it into bioavailable nitrogen, links their dynamics closely to the eutrophication problem of the Baltic Sea by adding nutrients to an already over-





fertilised ecosystem. Among researchers there is consensus that the fixed nitrogen is a major contribution to the overall nutrient budget. Quantitative estimates range from 20 to 50% of the total new nitrogen supply (as opposed to regenerated) that is available to the phytoplankton community (Adam et al., 2016; Gustafsson et al., 2017; Moisander et al., 2007; Vahtera et al., 2005; Whitton and Potts, 2002; Ploug et al., 2011). There is, however, no consensus on how nitrogen fixation will affect nutrient

loads in the Baltic Sea; even though a comprehensive understanding of this entanglement is key to an efficient environmental management of anthropogenic nutrient loads. One argument is that a reduction of the loads will have no net effect on the nutrient budget because cyanobacteria will compensate the reduction by fixing additional atmospheric nitrogen. A contrary view suggests that reduced loads will decrease primary productivity because nitrogen fixation is capped and cannot fully compensate reductions in nutrient loads. Among the reasons for a capped fixation are (1) limited availability of light which

throttles the metabolically expensive process of fixation, (2) limited bioavailability of one or several of the following elements: phosphorous, iron, molybdenum, (3) turbulent water movement (Moisander et al., 2002; Paerl and Huisman, 2009).

  Some of the numerous studies on cyanobacteria, which we will review in this study, are motivated by concerns to run into low-oxygen conditions; a risk that is apparently increasing because cyanobacteria favour the warming conditions that come along with global warming (Carey et al., 2012; Paerl and Huisman, 2009; Andersson et al., 2015b). The concern is that nitrogen

fixation may boost primary production which constitutes the source of organic material, sinking to depth. Subsequent reminer-alisation at depth by bacteria depletes oxygen concentrations and creates large-scale hypoxic or even anoxic environments that are lethal to fish and their fry (Elmgren, 2001; Elmgren and Larsson, 2001b; Nehring and Matthäus, 1991; Gustafsson, 2012; Diaz and Rosenberg, 2008, 1995; Breitburg et al., 2018).

  Further motivation to understand cyanobacteria dynamics in the Baltic Sea comes from some of the species' capability to

release toxins. There is evidence that the production of cyanobacterial toxins can increase with increasing nitrogen supply (Gobler et al., 2016; Dolman et al., 2012). Furthermore, it has been found in cultures that toxicity peaks when growth of the respective cyanobacteria is optimal (Lehtimäki et al., 1997). This is problematic because the toxins, can lead to mass-die-offs of mammals, fish and filtering organism (c.f. Breitburg et al., 2018; Sipiä et al., 2001; Karlsson et al., 2005; Paerl, 2014; Hense, 2007; Hense and Beckmann, 2010; Kuznetsov et al., 2008; Mazur-Marzec et al., 2013; Stal et al., 2003). In the Baltic one of

the most relevant species, *Nodularia spumigena* produces the toxin Nodularin. Additional thread comes from *Dolichospermum sp.* and *Aphanizomenon flosaquae*, who are also able to produce toxins. In addition to toxicity, intense cyanobacteria blooms can lead to a detrimental water clarity loss. In shallow coastal areas, this can shade benthic macrophytes, thereby effectively reducing their growth and survival, which in turn has negative effects on invertebrates and fishes, that use macrophytes as habitat for food and shelter (Short and Wyllie-Echeverria, 1996).

Despite the importance of cyanobacteria for the Baltic Sea ecosystem, the processes involved in the bloom formation of cyanobacteria are still not comprehensively understood (Hense and Beckmann, 2006). Numerous abiotic and biotic factors promoting cyanobacteria growth have been suggested and are often controversially discussed (e.g. Unger et al., 2013). Our present understanding of the dynamics of cyanobacteria, as summarised in biogeochemical ocean models, relies mainly on empirical field correlations rather than on a comprehensive understanding of physiological responses to environmental condi-





tions. These correlations, however, may well break under changing environmental conditions, thereby retarding model-based forecasts of the effects of potentially expensive management efforts.

In summary, deficient process understanding introduces considerable uncertainty to projections of numerical biogeochemical models, both globally (e.g. Landolfi et al., 2018) and in the Baltic Sea (e.g. Meier et al., 2012, their Fig. 7). Even so, such projections often support environmental management decisions - simply because superior alternatives are yet to be developed.

This study summarises knowledge about cyanobacteria dynamics in the Baltic Sea and compares this with the current state-of-the-art biogeochemical Baltic Sea models. Our aim is to identify knowledge gabs, thereby promoting the development of more reliable models. More specifically, we will (1) dissect the current generation of biogeochemical Baltic Sea models in an attempt to understand their underlying paradigms of cyanobacteria competitiveness and (2) review published studies focussed on observations and experimental results of, and controls on the most dominant cyanobacteria species in the Baltic:

*Aphanizomenon flos-aqua, Dolichospermum sp. and Nodularia spumigena* (Suikkanen et al., 2010).

We start with a comparison of five state-of-the-art model approaches in Section 2. Section 3 puts the models assumptions into the context of published observational and experimental studies. We will end with a discussion and summary in Sections 4 and 5, respectively.

## 2 Current Model Approaches

In the following we compare five coupled biogeochemical Baltic Sea models in terms of their mathematical formulations and underlying assumptions. Our choice of 5 is motivated by picking those that represent the state-of-the-art conveyed to stake holders, i.e. results from this class of models influence political decision making (c.f. Eilola et al., 2011; Meier et al., 2011, 2012, 2014; Neumann et al., 2012, 2002).

Four of the biogeochemical models, dissected here, are coupled to full ocean circulation models: *CEMBS* (Dzierzbicka-20 Głowacka et al., 2013; Nowicki et al., 2015, 2016), *ECOSMO II* (e.g. Daewel and Schrum, 2013, 2017), *ERGOM* (e.g., Janssen et al. (2004); Kremp et al. (2007); Kuznetsov et al. (2008); Neumann et al. (2002); Neumann and Schernewski (2005, 2008); Schernewski and Neumann (2005)) and *SCOBI* (e.g., Almroth-Rosell et al. (2011); Eilola et al. (2009); Meier et al. (2011b)). In addition, we include *BALTSEM* which, although it is a box-model rather than a full-fledged coupled ocean-circulation biogeochemical mode, has impacted stakeholders considerably ever since it has been developed in the early 1990s in an 25 attempt to support the Baltic Marine Environment Protection Commission - Helsinki Commission (HELCOM) and to develop the so called "HELCOM Baltic Sea Action Plan ( Gustafsson et al. (2017); Savchuk (2002); Savchuk (2012)). Specifically, we will review the reference version of *BALTSEM*, as initially developed for the Gulf of Riga (Savchuk, 2002).

### 2.1 General model structures

CEMBS, ECOSMO, ERGOM, SCOBI and BALTSEM are all so-called *mechanistic models* as opposed to statistical models.
They are, essentially, a set of partial differential equations that describe the temporal evolution of so-called *prognostic* entities of relevance or interest. Typical entities of relevance are variables such as nutrient, phytoplankton and zooplankton concentrations.





For each of these prognostic variables an equation is defined which puts their respective temporal derivative into relation with their biogeochemical sources and sinks which, typically, are interlaced. For example, the equation for phytoplankton comprises a sink term associated with zooplankton grazing. This sink term appears as a source in the zooplankton equation and thereby interlaces the zooplankton equation with the phytoplankton equation.

Complexity in mechanistic models, that are based on partial differential equations, is associated to the number of explicitly

resolved prognostic variables and the number of source and sinks terms for each of the variables. Conceptual problems arise because there is no consensus as concerns both the number of prognostic variable nor on the mathematical formulation of the respective source and sink terms. Typically, the respective parameters and formulations are rather *ad-hoc* choices which introduces substantial uncertainty to the realism of the model dynamics. An additional, albeit related, uncertainty is associated with the choice of *model parameters*: as a rule-of-thumb, each source and sink term necessitates at least one parameter. Such

parameters are typically not well constrained even though they determine the model behaviour in a fundamental way. Examples of these parameters are the maximum growth rate of phytoplankton and parameters which define the limiting effects of nutrient and light depleted conditions on autotrophic growth.

All models considered here are similar in that primary production fuelled by photosynthetically available radiation generates phytoplankton biomass which is proportional to the uptake of dissolved nutrients (BALTSEM being somewhat different here

because, by applying a variable Redfield ratio, it links nutrient uptake to carbon assimilation in a more flexible way than the other models investigated here). Explicitly resolved nutrients are nitrate, ammonium, phosphate (in all models considered here) and, optionally, silicate for diatoms (BALTSEM, CEMBS, ECOSMO only). The availability of nutrients in combination with light and (optionally) temperature determines phytoplankton growth rates.

It is common practice to group phytoplankton species into *functional groups* for each of which a distinct set of model param-

eters is defined. The models considered here are similar to one another in that they all differentiate between three functional groups with diatoms, cyanobacteria being common to all of them. As for the third group, ECOSMO, ERGOM and SCOBI refer to it as *flagellates*, BALTSEM as *summer species* and CEMBS as *small phytoplankton*.

A basic concept of the current generation of biogeochemical models is generally the widespread paradigm that diazotrophic cyanobacteria grow more slowly than ordinary phytoplankton and can, therefore, in most models only thrive when nitrogen is

no longer accessible to ordinary phytoplankton (LaRoche and Breitbarth, 2005; Paerl et al., 2016; Hense and Beckmann, 2006; Deutsch et al., 2007).

Sink terms which determine losses to phytoplankton abundances are designed to account for viral lysis, extracellular release and zooplankton grazing. All models considered here resolve one functional group of zooplankton, with the exception of ECOSMO-model, which resolves two (both, micro- and macro-zooplankton). As a general rule, the model parameters as-

sociated to zooplankton growth (fuelled by grazing on phytoplankton) are tuned such that top-down control terminates the phytoplankton spring bloom at the right time of the year. One may argue that the representation of zooplankton is more of a closure term than an attempt to realistically simulate zooplankton dynamics. In any case this approach calls for the definition of additional closure or sink terms for zooplankton. These sink terms for zooplankton (biomass) typically comprise the production





of fecal pellets, and death. Fecal pellets and dead zooplankton are the source of detritus (another prognostic variable) which sinks to depth where it is remineralised or lost to the sediment.

Typical attachments to the generic model backbone mapped out above, are additional prognostic variables such as oxygen and carbon (c.f. CEMBS and a later version of ERGOM (Kuznetsov and Neumann, 2013)) and a basic representations of the sediment. These are, however, beyond the scope of this review which focusses on cyanobacteria.

In the following we will elucidate differences among the models which, even though they share a similar basic structure, they can feature very different sensitivities to changing environmental conditions simply due to differing details in the specific formulations.

## 2.2 Growth formulation of Cyanobacteria

In this subsection, we compare the formulations of cyanobacteria growth. Special emphasis is given to the relations of
cyanobacteria growth to the respective other two functional phytoplankton groups, because these relations inherently defines the niche for cyanobacteria (i.e. the conditions under which they may outcompete other functional groups). Table 2 provides an overview for biogeochemical modellers in that it lists all model parameters considered in this study. In the following we will elaborate on the respective differences.

There is consensus among the models, that the growth of cyanobacteria is controlled by the availability of light, temperature
and phosphate. Most models assume that high temperatures accelerate the growth (Fig. 1a). The SCOBI and ECOSMO-models include an additional switch which shuts down cyanobacterial growth at salinities above 10 and below 11.5 psu, respectively. Yet another level of complexity is added in SCOBI where growth necessitates oxygen concentrations above 0.1 $mlO_2/l$ with growth gradually increasing above this oxygen threshold.

Moreover the model behaviour is imprinted by their respective mathematical formulations and several, often rather poorly
known, model parameters. All models considered here share the concept of a maximum growth rate that is multiplied by several other expressions that represent the external factors that limit this maximum growth capability. The actual maximum growth rates applied (c.f., Table 3) differ considerably among the models: in CEMBS (and ERGOM) the maximum growth of cyanobacteria is (less than) half compared to other models. Both models also assume that cyanobacteria grow, even at their maximum, rather slow compared to the other functional groups. In the other models, these differences are less pronounced.
Figure 1 puts this comparison into perspective, by accounting for the respective modulation by the water temperature. SCOBI and BALSEM, are strongly affected by water temperatures: cyanobacteria grow more than twice as fast at temperatures between 12-14°C than at temperatures below 12°C and growth accelerates further with rising temperatures (Fig. 1a). Above 15°C cyanobacteria grow almost as fast as diatoms (c.f. Fig. 1b). ERGOM does also account for the effects of ambient temperatures but the sensitivity is lower. For one, growth does never accelerate above the maximum growth rate of 0.5 $day^{-1}$ (thus, in
contrast to SCOBI, ERGOMs maximum growth rate is really a maximum rate. The ERGOM-model stalls all cyanobacteria growth below 12°C and sets maximum growth at temperatures exceeding ≈ .19°C. In contrast, the assumed increase in growth with temperature is rather gradual in the CEMBS model. The ECOSMO-model does not include any temperature dependence of phytoplankton growth. It does, however, include a strong light dependence of cyanobacteria growth which presumably has





a very similar effect because high light levels are typically related to high incoming solar radiation and shallow surface mixed layer depths which, in turn, are typically related to higher surface temperatures.

The maximum growth, as defined by the modulation of the maximum growth rate by the respective temperature sensitivity is damped under nutrient and/or light depleted conditions. The models under consideration differ considerably in this implementation. In SCOBI, the limitation of growth is implemented by multiplying with several factors all of which are smaller

than one. Each factor describes the limiting effect of one resource (such as phosphate concentration or availability of light). The other models apply the concept of Liebig, which assumes that the limitation is set by the most depleted essential resource rather than being the result of the combined effect of various depleted resources that potentially modulate one another.

All models considered here agree, that the growth of cyanobacteria depends on the availability of light and phosphate while other macro-nutrients are not limiting. Phosphate limitation is implemented by using a "Michaelis-Menten formulation"

(i.e., $\frac{PO_4^{3-}}{PO_4^{3-}+K_P}$). The respective *Half-saturation-constant*, $K_P$, varies substantially between models (Table 3), which imprints different sensitivities to phosphate limitation into their respective dynamics. Among the models, ERGOM is special in that it squares all terms (i.e. $\frac{(PO_4^{3-})^2}{(PO_4^{3-})^2+K_P^2}$) which steepens the nutrient-limitation-curve considerable, effectively setting a threshold rather than a gradual limitation.

Other than the steepness one consequence of differing $K_P$ is, that simulated cyanobacteria reach maximum growth at very

different levels of phosphate. SCOBI and ECOSM reach full growth already at very low phosphate concentrations, while CEMBS and ERGOM need much higher phosphate values to reach maximum growth - higher not only compared to the other models but also relative to their respective other functional groups.

Common to all models considered here is that the cyanobacteria are never limited by the availability of bioavailable nitrogen (one exception being cyanobacteria below the surface in ECOSMO). The fixation of nitrogen in the absence of ammonium

and nitrate is coupled to the uptake of phosphate in all models. Except for in BALTSEM, where the cyanobacteria top up their intracellular nitrogen concentrations until a predefined Redfield N:P ratio is reached. This predefined ratio is constant in all of the models with the exception of BALTSEM where it changes with water temperature, ambient N/P ratio and phosphate concentrations.

Despite the differences among the model formulations outlined above, all models agree in that cyanobacteria have an ad-

vantage over other functional groups under nitrate-depleted conditions - if phosphate is available. This phosphate, that has no Redfield equivalent of bioavailable nitrogen is also referred to as *excess phosphate*. In summary, all models agree in that excess phosphate is a necessary precondition for a cyanobacteria bloom.

Beside the impact of temperature and phosphorus limitation, there is consensus that the availability of light is essential to the growth of cyanobacteria. Details, however, differ: In BALTSEM, ERGOM and SCOBI light limitation ($light_{lim}$) for all

functional groups is expressed based on an assumed optimal light level ($I_{opt}$):

$$light_{lim} = I_{PAR}/I_{opt} * exp(1 - \frac{I_{PAR}}{I_{opt}}) \tag{1}$$

$I_{PAR}$ denotes the incoming *photosynthetically available radiation* (RAR) in the respective depth layer. $I_{opt}$ is set to 50 $W/m^2$ in BALSEM and 25 $W/m^2$ in SCOBI.





ECOSMO and CEMBS, in contrast, assume that the light requirement for cyanobacteria is higher than for other functional groups. This takes into account that nitrogen fixation is an energetically expensive process which has to break the dinitrogen molecule and thus has to overcome the strongest atom to atom bond among all bonds involving two atoms. In ECOSMO a threshold, exclusive to cyanobacteria, of $120\,W/m^2$ is, when undercut, shutting down all growth. In CEMBS the formulation of differing light sensitivities for the respective functional groups is more complex: built on a modification (which directly includes a calculation of self-shading effects) of the classical approach which expresses light limitation based a PI-curve ($light_{lim} = 1 - exp(-\alpha * PAR)$), CEMBS prescribes different initial slopes for each of the functional groups. The lowest initial slope, $\alpha =0.17$, is prescribed for cyanobacteria while the 0.3 and 0.34 for diatoms and small phytoplankton, respectively, imprint a higher competitiveness under low-light conditions. On a side note (which does not affect the competition between cyanobacteria and other functional groups in each of the respective models) is that the formulations, describing light attenuation within the water column, differ substantially among the models: BALTSEM accounts for shading effects of autotrophs, heterotrophs and detritus while ECOSMO only accounts for the autotrophs. In addition, the attenuation coefficients of photosynthetically available radiation in sea water itself vary almost by a factor of 3 from one model to another.

Indirectly related to the formulation of light limitation, is the representation of cyanobacteria's capabilities to control their buoyancy. Some species have gas vacuoles which give them the means to move upwards to the sun-lit surface. The respective model formulations, however, vary considerably: In the ERGOM model cyanobacteria have an advantage over other functional groups as they are positively buoyant. In the SCOBI the simulated cyanobacteria do not sink, while other phytoplankton does, while in BALTSEM the sinking speed of cyanobacteria is identical to other summer species (cf., Table 3).

### 2.3   Loss terms for cyanobacteria

Even more uncertain than the *s*ource terms, are the *s*inks of cyanobacteria. The models generally assume some phytoplankton mortality which can depend linearly and/or quadratic on the respective standing phytoplankton stocks. These fixed rates mimic complex processes, such as bacterial and viral lysis. In most considered models these parameters differ very little among functional groups (Table 4). Exceptions are BALTSEM and CEMBS: In the BALSEM-model the mortalities of the phytoplankton functional groups follow different temperature dependencies. In the CEMBS-model the mortalities differ among functional groups: cyanobacteria have a slightly higher linear mortality than other functional groups, while a quadratic phytoplankton mortality is set to zero only for cyanobacteria. Generally, these constant rates of phytoplankton mortality are set to rather small values and thus typically receive only little attention. Even so we want to note that this is of importance because the mortality determines the steady state solutions and, related, can drastically determine ecosystem responses to eutrophication (Löptien, 2011).

The largest loss term, however, is typically zooplankton grazing. The knowledge and process understanding of this component is still limited and the assumptions between models differ widely. Typically, this formulation is non-linear since the development of zooplankton biomass depends on its biomass at the foregoing time step. Another prerequisite for zooplankton growth or, rather, increase of its own biomass, is food availability. Here, the different models consider different potential food sources and use very different additional constraints: for example, the ECOSMO-model differentiates two zooplankton

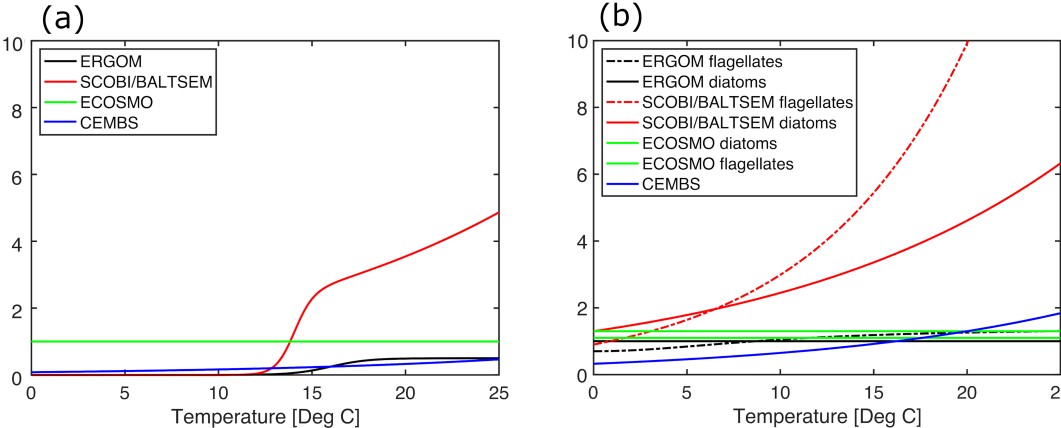

**Figure 1.** (a) Temperature dependence of the maximum growth for a) cyanobacteria and (b) for other functional groups in the five considered biogeochemical models $[day^{-1}]$.

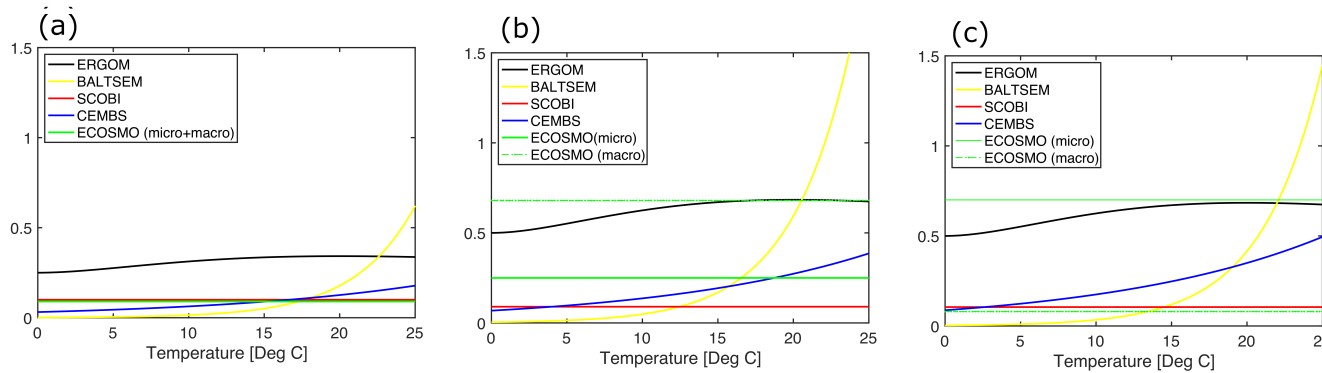

**Figure 2.** *Maximum grazing rates in dependence of temperatures for a) cyanobacteria b) diatoms and c) other phytoplankton $[day^{-1}]$. Note that the BALTSEM model prescribes zero grazing of diatoms in autumn. The values for the ECOSMO and the SCOBI-model refer to the product of max. zooplankton growth rate and food preference for the respective phytoplankton functional groups and the temperature effect. The two values for the ECOSMO-model refer to their two zooplankton groups: micro- and macro-zooplankton. All parameter values refer to the respective references provided in Table 4. Note, that the link to phytoplankton biomass differs considerably between the models and a direct comparison of the curves is not necessarily straight forward.*



groups (micro- and macro zooplankton) and assumes that micro-zooplankton feeds on phytoplankton, detritus and themselves, while macro zooplankton feeds additionally on micro zooplankton. In contrast, the ERGOM model assumes that their single explicitly represented zooplankton group feeds only on phytoplankton. In ERGOM, as in BALTSEM and CEMBS, maximum grazing rates are temperature dependent (Fig. 2), while SCOBI is the only model which assumes an oxygen dependency of the grazing rates. A comprehensive assessment of all these grazing formulations has not been published yet and is beyond the scope of this study. The major aim here is merely to compare those parameters and formulations which differ among functional groups within the models. In this respect selective grazing is of major interest. A comparison of the maximum grazing rates of the (competing) functional groups is shown in Table 4 and Figure 2. All models assume that zooplankton has a preference for phytoplankton and avoids cyanobacteria if possible. The magnitude of this preferential grazing is, however, unclear. Most models assume that the preference for cyanobacteria is by factor 2-3 lower than for other functional groups. ECOSMO is an exception with especially high food preferences. As a peculiarity, in the BALTSEM-model grazing rates depend additionally on the actual fixation rates, with higher nitrogen fixation rates reducing the grazing pressure on cyanobacteria.

## 3 Current knowledge about cyanobacteria's biology

### 3.1 Growth

The growth of cyanobacteria, like other photoautotrophic organism, depends on the availability of nutrients and light and is, additionally, influenced by other abiotic factors, such as temperature and salinity. In the following subsections we discuss factors impacting the growth of cyanobacteria and highlight the differences between cyanobacteria and other phytoplankton.

#### 3.1.1 Maximum growth

Generally, cyanobacteria are thought to have slow growth rates in comparison to eukaryotic phytoplankton cells (Butterwick et al., 2005; Hense and Beckmann, 2006; Lips and Lips, 2008; Rakko and Seppälä, 2014; Paerl and Otten, 2013; Paerl, 2014; Vahtera et al., 2005). This assumption is also reflected in the models (cf., Fig. 1). However, according to Reynolds (2006) and Foy (1980), who report rather similar maximal growth rates of cyanobacteria and micro-algae, this assumption has to be considered with some caution. Based on a study of 16 phytoplankton species (among others *Aphanizomenon flosaqua* and *Dolichospermum flosaqua*, which are rather common also in the Baltic Sea), Reynolds (2006) states that the maximal growth rates differ so substantially from species to species (even within functional groups) that generic statements about the functional groups cannot be made. The study by Foy (1980) explores 22 strains of 3 cyanobacteria genera (*Aphanizomenon, Dolichospermum sp.* and *Oscillatoria*). He draws the conclusion that algal size and shape appear to be better predictors of growth rates than the affiliation to a specific functional group. This statement is supported by the fact, that many cellular functions are strongly governed by the surface area to volume ratio (Kruk et al., 2010; Lewis , 1976). Further support showing similar results comes from Lürling et al. (2013), who tested eight different cyanobacteria and eight chlorophyte species (among





others *Aphanizomenon gracile* and *Dolichospermum sp.*). Table 5 and 6 summarise published maximum growth rates of the main cyanobacteria species in the Baltic Sea, along with those of a choice of ordinary phytoplankton species. Indeed, we find maximum growth rates differ substantially from on cyanobacteria species to another, and even among strains (Table 5). On average however, these studies report somewhat slower growth rates for the three cyanobacteria species than, for e.g. for chlorophytes (Table 5 and 6): For *Aphanizomenon flosaquae* maximal growth rates of 0.18-1.34 d$^{-1}$ are reported (Foy, 1980;

Gotham and Rhee, 1981; Rhee and Lederman, 1983; Konopka and Brock, 1978; Lee and Rhee, 1997, 1999; Sommer, 1981; Rakko and Seppäälä, 2014; Robarts and Zohary, 1987). *Dolichospermum sp.* show maximal growth rates of 0.4-1.27 d$^{-1}$ (Foy et al., 1976b; Konopka and Brock, 1978; Lürling et al., 2013; Nalewajko and Murphy, 2001; Reynolds, 2006; Oh et al., 1991; Sommer, 1981; Tang et al., 1997). Measured maximal growth rates of *Nodularia spumigena* range from 0.13 to 0.6 d$^{-1}$ (Cirés and Ballot, 2016; Lehtimäki et al., 1997; Sommer et al., 2006).

Maximum growth is only reached under optimal conditions that are rarely encountered in reality. In the following we explore the processes that inhibit maximum growth such as nutrient depletion, high light intensities and suboptimal temperatures.

### 3.1.2   Temperature dependence

Regarding temperature the model formulations differ widely and some models (e.g., SCOBI and BALTSEM) are based on the assumption that cyanobacteria require higher temperatures than ordinary phytoplankton to be able to grow optimally (Fig.

1). The respective model assumptions are roughly in agreement with Butterwick et al. (2005) who report that differences between species appear mainly at temperatures below 10°C and above 25°C, when testing the growth rate of 21 species within a temperature range between 2-35°C. Also, Foy (1980) report, that the temperature optima of cultured *Aphanizomenon flosaquae, Dolichospermum sp.* and others were similar to those of other planktonic autotrophs and Lürling et al. (2013) state that the growth rates of cyanobacteria and chlorophytes were very similar for a wide range of temperatures.

Regarding the typical Baltic Sea species, Table 7 shows optimal temperature ranges for the growth of the cyanobacteria species, compared to other phytoplankton species. Summarising the respective results, *Aphanizomenon flosaquae* has a somewhat wider optimal temperature range than *Nodularia spumigena*, spanning from 16°-31°C (Bugajev et al., 2015; Carey et al., 2012; Robarts and Zohary, 1987; Paerl and Otten, 2013; Degerholm et al., 2006). Temperatures which permit some growth are considerably lower. So report Cirés and Ballot (2016), that *Aphanizomenon flosaquae* is able to grow from 10°C. Üveges et al.

(2012) even measured intensive photosynthesis at 2-5 ° C. For *Dolichospermum sp.* optimal temperatures for maximal growth lie between 18° and 25°C, but shows slow growth starting from 10°C as reported by (Hellweger et al., 2016; Konopka and Brock, 1978; Robarts and Zohary, 1987; Paerl and Otten, 2013). *Nodularia spumigena* prefers relative high temperatures for optimal growth: 20-25*circ* :(Degerholm et al., 2006), while it is stated by Nordin et al. (1980), that *Nodularia spumigena* is still able to grow even around 5 ° C.

### 3.1.3   Nutrient demands

As diazotrophs are able to fix atmospheric nitrogen, bioavailable phosphorus is regarded as the essential limiting nutrient in the Baltic Sea for diazotrophic cyanobacteria (where iron is generally sufficiently available). Note, however, that N$_2$-fixation



is a very energy demanding process, which has the potential to reduce the growth rates by a factor up to 2 or 3 (Hense and Beckmann, 2006; Rhee and Lederman, 1983; Paerl et al., 2006) while too low phosphate concentrations can prevent any growth.

Phosphate (P) is essential, i.a., for cellular synthesis of nucleic acids, membrane phospholipids, as well as for energy transfer through tri- and bi-phosphorylated nucleotides (Degerholm et al., 2006). All models agree that the availability of phosphate is an essential precondition for cyanobacteria growth. The models do, however, differ considerably in terms of which P con-

centrations become limiting for the growth of cyanobacteria. The respective values for the half saturation constants envelope a huge range from 0.05 - 1.5 $mmolP/m^3$ (c.f., Table 3). Apart from the CEMBS model, the order of magnitude of the half saturation constants is comparable to other phytoplankton in all the models investigated here.

Similar to this laboratory experiments also show a huge range for the half saturation constant. Except for diatoms, the Table show that cyanobacteria can keep up with dinoflagellates and some green algae for P. Some cyanobacteria species have been

shown to feature a very high affinity for the uptake of P. This is because they are able to rapidly increase their P uptake rate by up-regulating two high-affinity P binding proteins and the phosphatase (Cottingham et al., 2015; Gobler et al., 2016). Additional adjustments to low P conditions include a reduction of their overall P requirements by substituting P-containing lipids with alternatives requiring less P. There is also evidence that cyanobacteria have relative low P requirements (Kononen and Leppänen, 1997; Degerholm et al., 2006). Even so Moisander et al. (2007) show in field- and laboratory experiments,

that P addition typically stimulates growth and nitrogen-fixation rates of *Aphanizomenon sp.* and *Nodularia spumigena*, which suggests that the availability of P is limiting in the Baltic Sea (despite the low P requirements). In this context Andersson et al. (2015a) found that Nostocales were dominating the cyanobacteria community under high total P and instead were negatively related to total N. Paerl et al. (2016b) reminds that $N_2$-fixing cyanobacteria dominate more often under N replete conditions, not under high P and low N conditions. So far there is no consensus, if the concentration of P or N or the relationship of N:P is

favouring dominance of cyanobacteria.

Again, the minimum P-requirements are species dependent: *Aphanizomenon flos-aqua* seems to have somewhat higher P demands and literature values for the half saturation constant range from 1-2.5 $mmolP/m^3$ (Degerholm et al., 2006; Gotham and Rhee, 1981; Healey at al., 1973). Olofsson et al. (2016) report, that in the Baltic Sea P concentrations of 0.04 $mmolP/m^3$ can limit *Nodularia spumigena* growth. Degerholm et al. (2006) indicate half saturation constants of 1-1.7 $mmolP/m^3$ for

*Nodularia spumigena* (Table 8). For comparison, e.g., Reynolds (2006) report much higher values up to 5.24 $mmolP/m^3$ for some green algae while the investigated diatoms seem to have rather low P-demands. The respective half saturation constants for diatoms in Table 8 range from 0.12-1.72 $mmolP/m^3$ (Fuhs, 1972; Werner, 1977; Holm and Armstrong, 1981; Tarutani and Yamamoto, 1994).

An interpretation of these facts is complicated by the ability of cyanobacteria to access dissolved organic phosphorus (DOP)

besides DIP. Phosphate ($PO_3^{4-}$) is the dominant form of DIP in natural waters and directly bioavailable for plants. In contrast, DOP is an integral part of the marine organic matter pool. DOP exists in a variety of forms which result from death and autolysis of organism, decomposition and excretion. Phosphorus esters (75%) and phosphonates (25%) are the two dominant forms of oceanic DOP (Clark et al., 1998). Not all phytoplankton possess alkaline phosphatase and can use DOP. From those,





who are able to use DOP, the efficiency of using DOP is very different. While f.e. dinoflagellates grow well under a variety

of DOP compounds, diatoms show a much lower and more restricted DOP utilisation (Wang et al., 2011). By being able to use a variety of DOP compounds cyanobacteria have a considerable advantage over other phytoplankton species, which are restricted to DIP or phosphorus esters (Degerholm et al. (2006); Dyhrman et al. (2006); Sohm and Capone (2006); Paerl (2014). Phytoplankton, who is able to use DOP, most commonly use alkaline phosphatase (AP) to hydrolyse phosphorus esters (Lin et al., 2012, 2016). Some phytoplankton species also have the potential to utilise phosphonates of the DOP compounds, which

are harder to extract for P (Dyhrman et al., 2006). This has been shown for many cyanobacteria species. They are able to access the more refractory phosphonates, as well as the semilabile phosphorus esters (Sohm et al., 2008; Orchard et al., 2010; Orcutt et al., 2013; Whitton at al., 1991; Dyhrman et al., 2006; O'Neil et al., 2012).

In accordance with this, various studies support an efficient use of DOP by cyanobacteria species of the Baltic Sea considered in this review. However, there are publications of conflicting results: Degerholm et al. (2006) show with their experimental work

that under limitation of DIP *Dolichospermum sp.* as well as *Nodularia spumigena* increase their alkaline phosphatase (APase) activity. This indicates that these two cyanobacteria species are able to use DOP for their growth. Degerholm et al. (2006) assumed, that *Nodularia spumigena's* high uptake ability for DOP, enables *Nodularia spumigena* to tolerate DIP-limitation during summer months. Studies of O'Neil et al. (2012) and Vahtera et al. (2007b) support the finding of Degerholm et al. (2006). However, in Vahtera et al. (2007b) experiments *Aphanizomenon sp.* was not able to use DOP. In strong contrast to

these findings is the work of Schoffelen et al. (2018). Their measurements contradict Vahtera et al. (2007b) by illustrating single-cell P-uptake rates with a nanoSIM (nanometer-scale secondary ion mass spectrometry). Their measurements show that *Aphanizomenon sp.* acquired only 15% of its P-demand from DIP and used instead about 85% from DOP. Whereas *Nodularia spumigena's* almost exclusively used DIP for its growth even at very low phosphate concentrations in their study. *Dolichospermum sp.* showed an intermediate behaviour. Note in this context, that models do generally not consider DOP as an

additional prognostic variable.

Another factor, which is rarely considered in models (an exception is BALTSEM), is the storage capacity of DIP by cyanobacteria. It is well known, that some cyanobacteria are able to drastically increase their intracellular P concentration (Nausch et al., 2008; Walve and Larsson, 2007, 2010; Sohm et al., 2011) and to store excess DIP. For the Baltic Sea this has been shown for *Aphanizomenon sp.* (Larsson et al., 2001). In May, after the spring bloom, *Aphanizomenon sp.* showed C:P

ratios around 50 ((Redfield, 1958), C:P=106:1). Nausch et al. (2009) even measured a C:P ratio of 32 after an upwelling event. However, Larsson et al. (2001) observed that during the subsequent build-up of biomass, P is used for growth and cellular P-concentrations are decreasing (C:P ratios around 400). Larsson et al. (2001) concludes that growth rate of *Aphanizomenon sp.* is limited by DIP availability in the Baltic Sea. Also, Raateoja et al. (2011) and Wasmund et al. (2005) share the opinion that, despite their DIP storage capacity, additional P sources are needed to sustain the today observed cyanobacterial blooms in

the Baltic Sea.





### 3.1.4 Light limitation

It is well known that cyanobacterial and algal growth rate is significantly influenced by the combination of light intensity and temperature (Butterwick et al., 2005). Under high light conditions, nitrogen-fixing cyanobacteria can capture more energy, can fix more nitrogen and by this increase their growth. High temperatures in combination with prolonged exposure to a high light intensity, on the other hand, may cause photoinhibition and induce harmful effects in algae (Ibelings, 1996). Photoinhibition also occurs when phytoplankton is shifted to irradiance substantially above those to which they have been acclimatised.

Cyanobacteria, however, can modify their photosynthetic apparatus within minutes to accommodate rapid fluctuations in light intensity or quality.

As differences between species are, again, substantial, we summarise the following studies sorted by species (see Table 9). For *Aphanizomenon flosaquae* optimum irradiance for photosynthesis are reported to be 6->33 $W/m^2$ (laboratory and field experiments)(Lehtimäki et al., 1997; Üveges et al., 2012). Photoinhibition for *Aphanizomenon flosaquae* was only found to

occur at light intensity of 99 $W/m^2$ (Pechar et al., 1987).

According to Eigemann et al. (2018) and Walsby and Booker (1980), *Dolichospermum sp.* prefers a rather low irradiance (3.72 - 8,32 $W/m^2$) (laboratory experiments and field experiments, based on the population peak at depth).

Several scientist tested the optimal light intensity for *Nodularia spumigena*. They state that *Nodularia spumigena* grows best at the highest light level tested (23 - 65.7 $W/m^2$), but can grow in a wide range of light intensities 5-114 $W/m^2$ (laboratory

experiments) (Eigemann et al., 2018; Jodlowska and Latala, 2019; Lehtimäki et al., 1997; Nordin et al., 1980; Roleda et al., 2008). Nordin et al. (1980) observe that very high temperatures (35° C) in combination with highest measured light levels (114 $W/m^2$) can prohibit the growth of *Nodularia spumigena*. Also, Jodlowska and Latala (2019) report reduced filament concentration and reduced photosynthesis rates at a combination of high light intensities and high temperatures (30° C). Noteworthy is, that Jodlowska and Latala (2019) did not find any photoinhibition of *Nodularia spumigena* until approximately

153 $W/m^2$ (700$\mu$ mol photons $m^{-2} s^{-1}$).

### 3.1.5 Buoyancy

Many cyanobacteria possess buoyancy regulation mechanisms, which enables them to actively control their position in the water column (Visser et al., 2016). Three of the respective control mechanisms are associated to internal buoyancy control: (1) they can modify the rate of gas vesicles being synthesised (Kromkamp and Konopka, 1986; Konopka et al., 1987), (2) they

can change their cell ballast by storage carbon which is produced by photosynthesis (mainly carbohydrates and proteins), (3) under prolonged irradiance and high photosynthesis rates the intracellular turgor pressure in some species can increase, which leads to the collapse of gas vesicle. An additional mechanism, related to Stoke's drag, is (4) colony size, with bigger colonies accelerating stronger in response to buoyancy changes.

Vertical mobility can be beneficial for cyanobacteria. E.g., it was suggested that cyanobacteria sink downwards to take up

nutrients from deeper waters (Cottingham et al., 2015; Konopka et al., 1987; Paerl, 1988) or to avoid harmful irradiance levels. Ganf and Oliver (1982) show, that cyanobacteria in shallow waters (*Dolichospermum spiroides, Microcystis aeruginosa*) are





able to migrate sufficiently up- and down the water column to reach nutrient rich water despite substantial density barriers. Also, cyanobacteria can migrate upwards or downwards to reach an optimal level of irradiance for photosynthesis (Whitton and Potts, 2002; Ibelings et al., 1991; Walsby et al., 1997; Walsby and Booker, 1980). Thus, vertical migration might give them a crucial advantage against their competitors. On top of the euphotic zone, cyanobacteria are able to maximise their photosynthesis and nitrogen fixation and additional they can shade their potential competitors. E.g. *Nodularia spumigena* is

adapted to very high irradiation levels and floats typically close to the water surface. Also, due to their buoyancy cyanobacteria don't face the risk to sink out of the euphotic zone like other phytoplankton.

The ability to migrate depends on size and morphology. Some cyanobacteria, like *Synechococcus sp.* (without gas vesicles) can only migrate a few cm per day - while others can float several meters. Large colonies with plenty of gas vesicles can undertake large diurnal distances in the vertical.

Below shown are the floating velocities separated by the studied cyanobacteria species of the Baltic Sea. Also, the listed studies are summarised in Table 9.

For *Aphanizomenon flosaquae* very different floating velocities of 3.5-51.8 m/d$^{-1}$ (Adam, 1999; Reynolds, 2006; Walsby et al., 1995) were measured. Walsby et al. (1995) also observed, that the floating velocities of *Aphanizomenon flosaquae* vary depending on light conditions, with higher velocities under low light.

In comparison *Dolichospermum sp.* is assumed to float much slower, as it does not form aggregates (Stal et al., 2003). Accordingly, Walsby et al. (1995) report respective floating velocities of 0.1-0.3 m/d$^{-1}$ for *Dolichospermum sp.*. Note in this context, that *Dolichospermum sp.* occurs mainly at a depth 5-10 m depth.

In the Baltic Sea *Nodularia spumigena* tends to float to the sea surface and there is no evidence for large changes in floating velocities under varying light conditions (Adam, 1999). Recorded floating velocities range from 35-45.7m/d$^{-1}$ for *Nodularia*

*spumigena* colonies (Adam, 1999; Walsby et al., 1995).

When considering the above numbers, however, one has to take into account that changes in buoyancy by gas vesicles is not a very fast process. The formation of gas vesicles takes generally more than a day (Oliver, 1994) and is a highly energetic process (Hense and Beckmann, 2006). It also requires nitrogen (Brookes et al., 2001; Klemer et al., 1982) and under phosphorus limitation the formation of gas vesicles is even slower (Konopka et al., 1987; Brookes et al., 2000). Some cyanobacteria possess

rather rigid gas vesicles. Species from the open Baltic Sea are observed to have relatively strong gas vesicles, while inshore populations have generally weaker gas vesicles. According to (Sellner, 1997), the strong gas vesicles of *Nodularia spumigena & Aphanizomenon flos-aqua* can even survive mixing down to 90 m depths. Cyanobacteria might, however, lose their buoyancy due to decreasing temperatures in autumn (Carey et al., 2012). Generally, colder water temperatures lead to a reduction in photosynthetic and respiratory rate and a much lower protein synthesis. Instead, glycogen is accumulated. This carbohydrate

ballast results in the sinking of cyanobacterial cells (Visser et al., 1995; Thomas and Walsby, 1986).

## 3.2   Loss Terms

There are several possibilities to die for cyanobacteria cells: cell death can be caused by necrosis by adverse environmental conditions, such as insufficient light, nutrients or temperature or by a programmed cell death (PCD) (Franklin, 2013). Cells can





be infected by fungi, undergo viral lysis (Munn, 2011) or can be grazed by a diverse selection of zooplankton (Franklin, 2013). The following subsections list the various causes for cyanobacteria loss. Note that the biogeochemical models investigated in

this study differentiate only grazing from other causes of cell losses. For all non-grazing related losses, the models generally assume a fixed loss rate which depends either linearly or quadratic on abundance (c.f., Sect.2.3).

### 3.2.1 Unfavourable Environmental Conditions

Under unfavourable environmental conditions cyanobacteria cells are subject to an increased cell loss: e.g. Sigee et al. (2007) report for *Microcystis flosaquae* at least 20%-50% senescent or dead cells by the end of summer. The necrosis of cyanobacteria

cells can be due to an injury response to harmful environmental conditions (e.g. high irradiance levels, salt stress, chemical perturbation, oxidative stress, phosphate limitation, iron limitation, high pH, low temperatures and low irradiance levels) (Lee and Rhee, 1999; Sigee et al., 2007). Among the ones confirmed in the Baltic Sea are mixing events, low irradiance levels, nutrient limitation and low temperatures (Vahtera et al., 2007a).

### 3.2.2 Programmed Cell Death (PCD)

In contrast to necrosis, cell death can result from an active physiological response of a cell in response to negative environmental conditions - a programmed cell death (PCD). There is growing evidence that PCD-like cell death can occur from unicellular amoebae and bacteria up to higher animals and plants (Ameisen, 1996). While the existence of PCD in cyanobacteria is still debated (Franklin, 2014), Claessen et al. (2014) describes that several lines of evidence point to the occurrence of PCD in filamentous cyanobacteria. E.g. individual cell death of *Microcystis flosaquae* occurs randomly throughout the colony and is

unrelated to any infection or cell cycle (Sigee et al., 2007). Instead, the cells switch actively to PCD. Similar *Trichodesmium sp.* and different strains of *Dolichospermum* display typical symptoms of PCD like morphological deformations, fragmentation and the subsequent autolysis of cells when exposed to stress (Berman-Frank et al., 2004; Ning et al., 2002). For *Dolichospermum flosaquae* it was found out, that cell death is controlled by circadian rhythms (Lee and Rhee, 1999), which implies that cell death is programmed in these organisms. Another hypothesis is, that PCD might be the result of asymmetric cell division

(Franklin, 2014). E.g. in *Dolichospermum solitarium* cell division always results in one large and one smaller cell. The larger cell act as a repository for metabolic waste, which will result into its early death, leading the smaller cell to a rejuvenation or increased fitness. All these studies indicates that PCD is part of the cyanobacterial life in the ocean and might be an important process in the decline of cyanobacteria blooms. Also, the cell death by PCD may facilitate biogeochemical cycling through transfer of organic and inorganic matter (Berman-Frank et al., 2004).

### 3.2.3 Infections and Lysis - direct Effects

Cyanobacteria cells can be infected by various organism, respectively entities, like fungi, bacteria or viruses. However, it is difficult to calculate the net effect of the different infections, because there are antagonistic effects between them.



### Fungi

Depending on the geographic region, the Baltic Sea resembles either a freshwater or a marine environment. The respective salinity threshold of fungi lies around 8 PSU. Depending on this classifications, the fungi community structure typically differs strongly. In salinities below 8 PSU fungal infections of bacteria and cyanobacteria cells are common. For instance, Mohamed et al. (2014) show that the fungi *Trichoderma citrinoviride* could lyse *Microcystis aeruginosa*. Similarly Kozik et al. (2019) report cell death of *Dolichospermum sp.* colonies due to infection by chytrid fungi.

### Bacteria

Many bloom-forming cyanobacteria like *Aphanizomenon sp.*, *Dolichospermum sp.* or *Microcystis sp.* are often closely associated with other microorganisms, especially heterotrophic bacteria (Gerphagnon et al., 2015; Liu et al., 2014). It has been reported that these prokaryotes exchange substances (organic matter, energy, oxygen, nitrogen, phosphate) (Levy and Jami,

2018), which lead to a better growth of both partners. This would be a mutalistic partnership. However, some heterotrophic bacteria instead use cyanobacteria cells as a food source (Gerphagnon et al., 2015; Hoppe, 1981; Paerl and Otten, 2013; Yamamoto et al., 1998) and in aged *Nodularia spumigena* filaments an immense number of bacteria has been observed. However, there is so far little evidence for strong lysing effects of bacteria on cyanobacteria in natural environments.

### Viruses

Viral lysis could have a much greater impact for cyanobacteria than bacterial lysis. Viruses are ubiquitous in aquatic environments. Estimates assume that virus concentrations are of more than $10^7$-$10^{10}$ viruses $ml^{-1}$ (Rohwer and Youle, 2010; Paerl and Otten, 2013; Fuhrman and Suttle, 1993; Šulčius and Holmfeldt, 2016; Suttle, 2005; Breitbart, 2012; Zeigler Allen et al., 2017). In the Baltic Sea, Ahrens (1971) reports up to $3.7 \cdot 10^7$ viruses per $l^{-1}$ for the bacteria group agrobacterium (which

covers only a fraction of all bacteria). The more recent studies by Riemann et al. (2009) and Holmfeldt at al. (2010) estimate virus concentrations in a range from $3.0 - 4.9 \cdot 10^7$ viruses per $ml^{-1}$. (For reference, concentrations of viruses in air (outdoor) are around $36 \cdot 10^1$ viruses per per $l^{-1}$ (Prussin et al., 2015)).

Viruses are the most abundant biological entities in the ocean and their numbers can exceed that of bacteria by about 5-10 fold (Silveira and Rohwer, 2016; Paerl and Otten, 2013). Most of these viruses are bacteriophages (a virus that infects and replicates

within bacteria and archaea). Cyanophages are viruses that infects and replicates within a cyanobacteria. Viruses follow a similar distribution pattern like bacteria: with higher abundances in productive waters and lower abundances in oligotrophic regions (Bratbak et al., 1994; Fuhrman and Suttle, 1993). Also, viruses are much more abundant in freshwater than in marine systems (Maranger and Bird, 1995). In the Baltic Sea, virus-induced bacterial mortality is among the highest reported for temperate aquatic ecosystems (Weinbauer and Rassoulzadegan, 2003).

The importance of viruses in controlling cyanobacteria abundances is, however, poorly understood. There are only few and mostly observational evidences for the impact of viruses on cyanobacteria. A wide range of 20-50% has been estimated for daily marine bacteria cell mortality due to viral lysis (Suttle, 2005; Fuhrman, 1999; Breitbart et al., 2007). Other studies





indicate, that viral induced bacterial mortality might be an important factor, controlling bacterial, algal and cyanobacterial abundances (Wommack and Colwell, 2000; Bratbak et al., 1993; Fuhrman and Suttle, 1993; Šulčius and Holmfeldt, 2016). This mortality most likely exceeds the effect of zooplankton grazing, especially in nutrient-rich, brackish waters like the Baltic Sea (Paerl and Otten, 2013). In a mesocosm experiment in the Baltic Sea, Bratbak et al. (1992) estimate that viral lysis leads to a reduction of up to 72% of cyanobacterial cells per day. However, in Australia it was shown, that cyanobacteria cells, which survive a virus attack, later develop a resistance against the virus (Tucker and Pollard, 2005). Still Šulčius et al. (2015) could

show in the Baltic Sea, that growth rate of *Aphanizomenon flosaquae* was significantly suppressed by a virus infection. While lytic infections (viral reproduction cycle which results in the destruction of the infected cell) do not play an important role when diversity of bacteria/ cyanobacteria is high, a bloom of one specific host increase the risk for lytic infections drastically ("killing the winner hypothesis")(Bratbak et al., 1994). In enclosure experiments, Simis et al. (2005) could indeed observe the termination of cyanobacteria blooms due to virus pathogens.

In summary, ongoing research indicates that viruses could play a very important role in controlling cyanobacteria blooms (Weinbauer at al., 2003; Tucker and Pollard, 2005). However, there are missing precise quantitative studies to really assess the influence of viruses on cyanobacteria in the Baltic Sea.

**Infections and Lysis - antagonistic, indirect Effects**

High cell turnover, due to virus lysis, has a large impact on daily nutrient recycling. Especially bacteria and flagellates benefit from the released nutrients from lysed cyanobacteria cells. Even cyanobacteria themselves are, most likely, able to recycle the nutrients from lysed cells (Breitbart, 2012; Bratbak et al., 1994; Šulčius and Holmfeldt, 2016; Hewson et al., 2004). Bratbak et al. (1992) could show in a mesocosm experiment, that the whole bacterial and primary community could be sustained with organic phosphorus released from lysed cells. Especially in the Baltic Sea for *Nodularia spumigena*, with its low phosphorus

requirements and high affinity for DOP (Degerholm et al., 2006), recycled phosphorus might be an important continuous nutrient source.

Another aspect of viruses is, that viral lysis can shorten long filaments of cyanobacterial blooms. Šulčiu et al. (2017a) observed that viral lysis changed the filament morphology of a cyanobacteria colony substantially. During the experiment the mean filament length of the Baltic Sea species *Aphanizomenon flosaquae* decreased by 58%, which made the cyanobacteria more

vulnerable to grazing by zooplankton. Also, cyanobacteria have been observed to produce an exopolysaccharide matrix that surrounds the colony-embedded filaments as protection against grazing and viral lysis (Šulčiu et al., 2017b).

### 3.2.4   Grazing on cyanobacteria

Several studies suggested that there is hardly any grazing on cyanobacteria due to their toxicity, bioactive compounds that

hamper digestion, bad taste, poor content of lipids, large filamentous size and low food quality (Carey et al., 2012; Daewel and Schrum, 2013; Ger et al., 2016). Generally, prey morphology is thought to be one of the most important factors influencing zooplankton grazing (Gerphagnon et al., 2015). With their large, filamentous size, most bloom forming cyanobacteria in the





Baltic Sea are difficult to graze. The toxins produced by cyanobacteria are another obstacle for grazers as these can be lethal. E.g. microcystin is lethal for a wide range of daphnia and copepod species (DeMott and Moxter, 1991; Ger et al., 2019). Ciliates, however, seem to be the most tolerant grazers to microcystin.

Despite all these obstacles there is grazing on cyanobacteria. One way is that grazers can develop a certain tolerance: especially during longer exposure to cyanobacteria, there can be a remarkable high biomass of small-bodied zooplankton which co-exist with cyanobacteria (Bouvy et al., 2001; Davis et al., 2012; Ger et al., 2016). This coexisting can lead to

better adapted grazer species (Davis et al., 2012; Bouvy et al., 2001; Sousa et al., 2008). DeMott and Moxter (1991) found, that copepods & daphnias became more tolerant against toxins after exposure to toxic cyanobacteria - or better in avoiding them. This tolerance could even be transferred to their offspring (Gustafsson et al., 2005). Sarnelle (2007) observed that high abundances of generalist may even control cyanobacteria blooms, if zooplanctivorous fish are rare. Carpenter (1989) proposed, that cyanobacteria may be highly vulnerable to grazing at the time of initial recruitment. Inline, Chan (2004, 2006) could show,

that zooplankton can suppress cyanobacteria blooms.

However, while there is some grazing on cyanobacteria and grazing might be important during the start of the growing phase and during termination of cyanobacteria blooms, worldwide and specifically in the Baltic Sea it can be observed, that under favourable conditions, cyanobacteria can outgrow any grazing pressure easily (Walve and Larsson, 2007; Ger et al., 2016; Sellner, 1997; Paerl and Otten, 2013). This rather low grazing pressure on cyanobacteria has a huge ecosystem impact, as,

consequently, during cyanobacteria blooms a lower proportion of the primary production is consumed by larger grazers and therefore not transferred to higher trophic levels. These findings are represented in the models by assuming that grazing on diatoms or other phytoplankton is 2-4 times higher than grazing on cyanobacteria. There is, however, no consensus yet on the exact formulation of zooplankton grazing in the current model generation and grazers are represented by a single (BALTSEM, ERGOM, SCOBI and CEMBS) or two (ECOSMO) functional groups.

In the Baltic Sea a wide range of possible grazers on cyanobacteria exist. To highlight the differences between different grazers, we discuss in the following the most abundant grazers (copepods, rotifers and cladocerans) and some grazers, which are known to be able to exert a strong impact on cyanobacteria in other ecosystems (bivalves, protozoa). In each group their occurrence in the Baltic Sea, their feeding mode and the known grazing pressure on cyanobacteria are presented.

**Copepods**

Among the most important and abundant grazers in the Baltic Sea are copepods (e.g. *Acartia logiremis*, *Temora longicornis* or *Centropages hamatusarexs* are common in the Baltic Sea). Copepods are a group of small crustaceans and belong to the mesozooplankton. They are able to select their food and can avoid cyanobacteria or select smaller colonies of cyanobacteria

while grazing Ger et al. (2019). One reason for their avoidance of cyanobacteria are the cyanobacterial toxins, which are lethal for many copepod species. Also many filamentous cyanobacteria are too large to be grazed on. In accordance with this, most studies about copepod grazing on cyanobacteria in the Baltic Sea show their avoidance and no significant grazing effect on





cyanobacteria (Sellner et al., 1994, 1996; Engström, 2000; Sommer et al., 2006). On the contrary, copepod-dominated zoo-plankton communities may also facilitate cyanobacteria by praying selectively on the eukaryotic competitors (Ger et al., 2016; Hong et al., 2013). Similar results are found by Eglite et al. (2019), who could demonstrate based on fatty acids-, amino acids- and stable carbon isotopes analysis, that mesozooplankton obtained essential fatty acids (FAs) and amino acids (AAs) from cyanobacteria via feeding on mixo- and heterotrophic (dino-)flagellates and detrital complexes. Overall, while copepods play

an important grazing role in the ecosystem, they will not be able to control cyanobacteria growth in the Baltic Sea (Sommer et al., 2006).

### Cladocera

The cladocera are another group of small crustaceans which range in size from 0.2-3.0 mm (with some exceptions). While

most cladoceran species live in fresh water, eight species are truly oceanic. In the Baltic Sea the cladoceras *Bosmina coregoni*, *Evadne nordmanni* and *Daphnia cristata* are very common among others.

Cladoceran are generalists, but there is a profound difference in the methods of food collection between littoral, planktonic and predatory cladocera species (Smirnov, 2017). Planktonic species are mostly filter feeders. However, members of the family Bosmina are known to have a dual feeding mechanism. They have mesh-like structures for filtering but they also can grab larger

particles with their first two thorac limbs (Bleiwas and Stokes, 1985). By this, they are able to select for food items. In line with this, Kerfoot and Kirk (1991) demonstrate that two species of *Bosmina spp.* consume algal foods by size and taste. They show preferences for small algae, bacteria and detritus (Solis et al., 2018). In contrast to this, filter feeding cladoceras consume algae that are present in the water, as well as organic particles and bacteria. Due to their feeding method, they are not able to avoid toxic cyanobacteria (Ger et al., 2019). Still, studies of Ismail et al. (2019) report, that cladoceras species (here Daphniidae)

were mostly feeding on small green algae cells. Filamentous cyanobacteria (*Dolichospermum circinalis, Microcystis flos-aqua* and *Dolichospermum sp.*) were also utilised by the grazers, but to a minor degree. In contrast to this is, that in freshwater lakes cladoceras are among the most important grazers on cyanobacteria. Especially the large-bodied *Daphnia magna* (max. female size 5 mm) is capable of suppressing filamentous cyanobacteria (Urrutia-Cordero et al., 2016). Whereas *Daphnia cristata* is one of the smallest daphnia (max. female size 1.6 mm) and cannot exceed high grazing pressure on filamentous cyanobacteria

and *Bosmina coregoni* and *Evadne nordmanni* have a similar size (max. female size 1.5 mm). In summary, in the Baltic Sea there is an abundant number of cladocerans, mostly smaller species. In contrast to freshwater habitats, studies suggest, that cladocerans will not be able to suppress cyanobacteria blooms.

### Rotifera

Another important grazing group are rotifers. Typical species in the Baltic Sea are among others *Synchaeta baltica* and *Keratella quadrata*. They are much smaller zooplankton species than copepods and belong to the size class micro-zooplankton. However, since the rotifers have higher metabolic rates and are considerably more abundant than crustaceans, they may be important in the structuring of plankton communities (Gilbert and Bogdan, 2017). Also, rotifers have the fastest reproductive





rates of any metazoans (Mironova et al., 2008) and can reproduce unisexual or bisexual. Therefore rotifers can quickly respond to altered food supply. In the brackish waters of the Baltic Sea rotifers are a highly diverse and widely distributed group. They are especially diverse and abundant (up to 95% of zooplankton biomass) in coastal ecosystems (Ojaveer, 2010). With increasing salinity, abundance and diversity of rotifers decrease, due to the freshwater origin of this group. Furthermore, in the open

Baltic Sea, rotifers are less diverse than in the Baltic Sea estuaries (Mironova et al., 2008). Most rotifers are suspension-feeders. Due to their small size, their diet must be tiny as well. It mostly consists of dead or decomposing organic material as well as unicellular microalgae, bacteria or protozoans (Mironova et al., 2008). Some species are known to be cannibalistic. Rotifers will not graze on living filamentous cyanobacteria, due to their size. However, decaying cyanobacteria or smaller cyanobacteria like 'Synechococcus' can be a target for rotifers.

**Bivalves**

In general bivalves are filter feeders. However, at least some mussel species can distinguish between food particles and show different clearance rates for different particles sizes or particles types. Another way how mussels are selecting food items is to reject particles by discarding them into pseudofeces (Ward et al., 2004; Tang et al., 2014). Lesser et al. (1991) report, that

e.g. scallops showed lower clearance rates for toxic dinoflagellates then for three different plankton species. Regardless of the partial ability to select food particles, mussels are very successful grazers on cyanobacteria in freshwater lakes. E.g. population of *Dreissena spp.* were able to reduce cyanobacteria in several occasions (White at al., 2014) and even prevented or terminated cyanobacteria blooms (Gulati et al., 2008; Baker et al., 1998). In the Baltic Sea, due to its brackish water, *Dreissena polymorpha* has become a part of the Baltic coastal ecosystem in many areas. While its distribution is patchy along the coast of the

Baltic Sea (Werner et al., 2012), in the oligohaline southern and eastern coastal lagoons and inlets of the Baltic Sea *Dreissena polymorpha* is one of the most common species (Snoeijs-Leijonmalm et al., 2017) and could play a role in decreasing cyanobacteria blooms in coastal areas.

**Protozoa**

Today we know that protozoans, such as ciliates and flagellates, are an important group of grazers on cyanobacteria (Worden et al., 2015). Due to their small size they mainly consume unicellular cyanobacteria. Some ciliates and amoebae however, can engulf prey items several times their own length by breaking down trichomes or encapsulating individual cells from cyanobacterial colonies (Ger et al., 2016). Dryden and Wright (1987) gave a great overview of grazing on cyanobacteria by different protozoa classes. The cited studies (mostly laboratory experiments), showed dinoflagellates, amoebae and ciliates

grazing on the order Nostocales, which comprises the here studied cyanobacteria. E.g. Hoppe (1981) measures in the Baltic Sea, that the ciliate *Nocardia sp.* become the dominant species in aged cyanobacteria filaments. *Nocardia sp.* can cause lysis of cyanobacteria (Hoppe, 1981). In lakes Canter et al. (1990) observed, that the cyanobacteria population crashed under grazing pressure of the ciliate *Nassula spp.*. *Nassula spp.* has the remarkable ability to ingest long filaments of cyanobacteria by sucking them in, spaghetti-like and coiling them intracellularly (Reynolds, 2006). Similar results reported Boyer et al. (2011), that while





microzooplankton (among others ciliates) community did not graze on filamentous cyanobacteria during the summer bloom, by autumn, as the cyanobacteria bloom was declining, microzooplankton grazing rates were high particular on *Aphanizomenon flosaquae*. Cook (1976) report that the amoebae *Mayorella* were actively and exclusively grazing on *Dolichospermum* cells, reducing within 3 days the cyanobacteria bloom into a milklike surface film. Also, Reynolds and Walsby (1975) observed a

rapid collapse of a *Dolichospermum circinalis* bloom after being attack by a large population of ciliates *Ophryoglena atra*.

These studies show, that protozoans have the potential to exceed a strong grazing pressure on cyanobacteria. In the Baltic Sea the above mentioned species *Mayorella sp., Ophryoglena sp. and Nassula sp.* do occur. Their importance still has to be examined.

### 3.3   Others

#### 3.3.1   Salinity constrains

The Baltic Sea features a wide range of salinities, ranging from 15-25 PSU in the northwestern-most part of the Baltic to 2-3 PSU in the Bothnian Bay. The Baltic Proper, situated in the centre, is characterised by intermediate values around 6-8 PSU. The large spatial variance in salinity can induce large local salinity variations over time when ocean currents mix water masses

from different origins. This can decrease the local growth- and photosynthesis rates of algae and cyanobacteria, once specific salinity thresholds are over- or undercut and physiological stress sets in. By this means-salinity has the potential to control the occurrence of cyanobacteria species. Salinity thresholds are set in SCOBI and ECOSMO, while the other models do not include salinity constrains on simulated cyanobacteria. The respective assumptions in SCOBI and ECOSMO refer to regions with extreme salinities - both extremely high and extremely low - where cyanobacteria growth is set to zero.

Field observations show that in most parts of the Baltic Sea large cyanobacteria blooms occur during summer, except in the relatively saline waters in the Kattegat and the Belt Sea. Thus, e.g., Rakko and Seppäälä (2014) conclude that high salinities seem to restrict the growth of Baltic Sea cyanobacteria and estimate a threshold around 10 PSU. Low salinities, in contrast, do not seem to restrict the growth of cyanobacteria in the Baltic Sea in general and several studies report blooms at very low values: Wasmund (1997) relates Baltic Sea blooms to salinities between 3.8-11.5 PSU and Kononen et al. (1996) report no

significant correlation of the bloom-forming cyanobacteria in the Gulf of Finland with salinity (i.e., salinities between 3-6 PSU).

The differences between species are substantial. E.g., Lehtimäki et al. (1997) state that in the northern part of the Gulf of Bothnia, where the water mass properties are close to freshwater, *Nodularia spumigena* is absent. Brutemark et al. (2015) even stated, that salinity might be one of the main factors that explains the distribution of species in the Baltic Sea.

These statements are supported by laboratory experiments which we list in the following for the most relevant species.

*Aphanizomenon flosaquae* is well known as a freshwater specie (Laamanen et al., 2002). Accordingly, Rakko and Seppäälä (2014) and Laamanen et al. (2002)measured rather low salinities of 0-5 PSU for optimal growth. Rakko and Seppäälä (2014) describe this species as rather coastal, preferring less saline conditions. Inline, Lehtimäki et al. (1997) stated that *Aph-*





*anizomenon flosaquae* is not able to tolerate salinities higher than 10 PSU. Consistently, its abundance decreases from the northern to the southern part of the Baltic proper.

The taxa *Dolichospermum* originate from freshwater, with some strains adapted to brackish water (Brutemark et al., 2015). In agreement, in the Baltic Sea the specie *Dolichospermum flosaquae* shows similar growth rates between 0-10 PSU and a strong decrease in growth rates at higher salinities (Moisander et al., 2002).

For the species *Dolichospermum aphanizomenoides*, Moisander et al. (2002) report a wide range of salinities, from 0-10 PSU, which are related to similar growth rates. *Dolichospermum* taxa originate from freshwater, with some strains that are adapted to brackish water (Brutemark et al., 2015). In the Baltic Sea *Dolichospermum spp.* can grow in salinities between 0-10 PSU, but their growth rates decrease strongly above 10 PSU (Moisander et al., 2002).

For different strains of *Nodularia spumigena* a wide range of tolerable salinities were reported: the widest reported range is 0-20 PSU (Moisander et al., 2002), while Rakko and Seppälä (2014) narrows the optimal salinity range down to 8-10 PSU. Studies of (Lehtimäki et al., 1997), Mazur-Marzec et al. (2005) and Nordin et al. (1980) find salinity ranges in between (5-18 PSU). Results by Moisander et al. (2002) suggest that the actual salinity range may vary from strain to strain and we speculate that this explains the differing findings. (Mazur-Marzec et al., 2005) also observed a significantly inhibited growth at extreme salinities of 0 and 35 PSU. Interestingly, salinity concentration apparently affect the toxicity of cyanobacteria blooms: Mazur-Marzec et al. (2005) report an increase of Nodularin production for *Nodularia spumigena* with increasing salinity concentrations. Inline, Brutemark et al. (2015) found the highest intracellular toxin concentration at the highest tested salinity concentrations (6 PSU) for *Dolichospermum spp.*.

To sum up, in most areas of the Baltic Sea, salinity is no restriction for the growth of cyanobacteria.

### 3.3.2 Cyanobacteria akinetes

The considered biogeochemical models do not consider the life cycle of cyanobacteria (e.g., the resting spores *akinetes*). A respective approach, was, however, tested successfully in a low-dimensional setup by Hense (2007) and Hense and Beckmann (2010). In the Baltic, all dominant bloom forming diazotroph cyanobacteria taxa are able to produce akinetes. These resting spores have a slightly thicker cell wall than vegetative cells and are more resistant to environmental stress (e.g. temperature, desiccation) (Agrawal, 2009; Paerl, 2014; Kaplan-Levy et al., 2010). Some akinetes will germinate shortly after formation or after a more or less longer maturation period. For example, akinetes of *Dolichospermum sp.* were able to germinate even after a dormant phase of 64 years (Kaplan-Levy et al., 2010). Laboratory experiments indicate, that akinetes need a certain time to mature and to reach a certain energy or nutrient level, before germination is initiated again (Hense and Beckmann, 2006; Kaplan-Levy et al., 2010). The dormant akinetes can act as over-wintering stadium or they can ensure the long-term survival of the population. Different stimuli like temperature, light intensity, a low DIN/DIP ratio, sediment resuspension, salinity can trigger germination of akinetes (Karlsson-Elfgren et al., 2004; Rengefors et al., 2004; Kaplan-Levy et al., 2010; Huber, 1985; Sommer et al., 2006).



Field observations and mesocosm experiments indicate, that the importance of akinetes for the main bloom-forming cyanobacteria taxa in the Baltic Sea differs from species to species (Sellner, 1997; Agrawal, 2009; Rother and Fay, 1977; Suikkanen et al., 2010). For example: *Aphanizomenon flosaquae* seems to have a holoplanktonic life strategy (Rother and Fay, 1977; Jones, 1979; Head at al., 1999; Wasmund, 2017). It can produce akinetes, but during winter a "refuge population" of filaments can be observed in deeper waters (Heiskanen and Olli, 1996) even under sea ice (Laamanen et al., 2002), from which in spring/ early summer the population will develop (Sellner, 1997; Suikkanen et al., 2010). Observations for *Dolichospermum spp.* are not conclusive: In the Baltic Sea, akinete formation of *Dolichospermum spp.* has been documented (Olli et al., 2005). However, Rother and Fay (1977) observed that the bulk of akinetes of *Dolichospermum spp.* germinated shortly after sporulation and that the over-wintering population consist of vegetative filaments, while Suikkanen et al. (2010) stated that *Dolichospermum spp.* regularly germinated from overwintering akinetes in the sediment. Cirés et al. (2013) found a dual strategy for *Dolichospermum spp.*, with major akinete production combined with pelagic overwintering filaments. *Dolichospermum spp.* is the least dominant taxa of these three main bloom forming Nostocales in the Baltic Sea. The other major bloom forming taxa in the Baltic Sea: *Nodularia spumigena* - seems to have a mixed life strategy. It produces akinetes that overwinter in shallow sediments (Jones at al., 1994), but *Nodularia spumigena* only germinated from akinetes occasionally and mainly uses trichomes that overwinter in the water column (Suikkanen et al., 2010). In tank experiments *Nodularia spumigena* only grew within water from above the halocline, without additional input from akinetes (Wasmund, 2017).

In summary, the effect of akinetes on the extent of subsequent blooms, however, is still under discussion and difficult to estimate. Based on field observations and literature data Kaplan-Levy et al. (2010) draw the conclusion that the contribution of akinetes towards the bloom success of next year's population of Nostocales seems to be rather small. Cirés et al. (2013) suggest, that the growth rate of the initial population has a bigger impact on bloom formation than the size of the inoculum (Rücker et al., 2009). Therefore, (Kanoshina et al., 2003) assume that beneficial environmental conditions at the start of the growing season are the major factor for the development of extensive cyanobacteria blooms.

## 4 Discussion

In the foregoing chapters, we provided an overview over the current knowledge about Baltic Sea cyanobacteria from the perspective of both, modellers and biologists. In the following we discuss key differences between model approaches and observational evidences (Sect. 4.1). Sect. 4.2 debates the impact of the oceanic processes to the Baltic Sea, respectively. Some considerations about model assessment metrics are in sect. 4.3.

### 4.1 Biogeochemical processes

We compared five state-of-the-art biogeochemical models, designed for political decision making regarding their underlaying assumptions for simulating cyanobacteria abundance. Our elaborations illustrate that, to date, there is only limited consensus on the degree of simplification and concerning key processes necessary for a reliable simulation cyanobacteria distribution and biomass. There is a certain agreement between modellers to focus only on one cyanobacteria class whose growth's is generally





controlled by the availability of light and phosphate. All model parameters associated to this single class of cyanobacteria is necessarily a compromise of the very different observed values for different species (cf., Table 5, 7 and 8). This compromise is typically reached by adjusting the model parameters until the simulated cyanobacteria show a reasonable agreement with observations (c.f., Löptien and Dietze (2015), Schartau et al. (2017)). Somewhat disconcerting the respective parameter choices differ substantially from one model to another (Table 3 and 4; Fig. 1 and 2). To this end we regard particularly the various choices of growth permitting phosphate thresholds as critical, because we suspect that these parameter values might have a drastic effect on model based investigations of nutrient load scenarios. Another potentially problematic assumption which introduces substantial uncertainty in model simulation is the fixed Redfield-ratio engrained in most models. For projections into our warming future we suspect that especially the very different temperature dependencies introduce substantial uncertainty in the model responses. An additional aspect introducing uncertainty is the need to distinguish between dissolved inorganic (DIP) and dissolved organic phosphorus (DOP) in models. Another open question is the need to include nutrient storage capacities into the models. Largest uncertainties in the model formulations, however, regard the bloom termination. The mechanisms are in general not well understood and require further research. Apart from grazing, viruses were shown to be potentially very important. Still, the respective knowledge is today too limited to make pertinent model assumptions.

## 4.2 Abiotic oceanic processes

The occurrence of cyanobacteria is controlled by several abiotic factors, such as nutrient availability, light, temperature, mixed layer depth, currents and upwelling. For example, the diazotrophic species *Aphanizomenon sp.* is able to utilise the upwelled DIP, despite relative cold temperatures in the upwelled waters (Nausch et al., 2009; Lips and Lips, 2008). Therefore, as up-welling events will affect the abundance of cyanobacteria, a correct representation in the underlying ocean model is of particular importance. To capture the latter, the respective ocean models need a sufficient spatial resolution and also reliable boundary conditions and the respective derivation of the surface fluxes are of major importance (e.g., Dietze and Löptien (2016)). Without a good representation of these factors in the underlying ocean model parameter adjustments in the biogeochemical component lead inevitably to *reciprocal bias compensation*, i.e. flaws of the ocean component are compensating by tuning the biogeo-chemical module to obtain a reasonable good agreement between model and observations. The problem with such an approach is that such a bias compensation is not necessarily robust under changing environmental conditions (Löptien and Dietze, 2019).

## 4.3 Model assessment metrics

Given the above considerations, the assessment of a model, that is designed for sensitivity studies, future projections and political decision making cannot be limited on cyanobacteria alone. Rather, a meaningful model assessment needs to take additionally the controlling mechanisms into account (such as the representation of nutrients, light, mixing, upwelling). So far, however, no common model evaluation criteria exist. A major difficulty in this respect is the still limited data availability. It is already challenging to obtain large-scale observations of cyanobacteria, let alone all other above mentioned data (e.g. light conditions, nutrient distributions, zooplankton concentration and virus abundances). Today, satellite data are, due to their easy accessibility, often the first choice. While it is well known, that these data contain a considerable degree of uncertainty (e.g.





Reinart and Kutser, 2006), it is the only data source to assess large scale pattern of cyanobacteria and to evaluate simulated
cyanobacteria blooms Baltic-wide. Thus, several studies compare the simulated distribution of cyanobacteria with satellite
images. However, so far these evaluations are mostly limited to long-term averages or interannual variations of the maximum
or cumulative surface area covered by cyanobacteria in the Baltic Proper (e.g. Eilola et al., 2011; Janssen et al., 2004). On a
positive note, the amount of available in-situ data increases continuously (e.g., by Ferry boxes: Kaitala et al. (2014); Łysiak-
35 Pastuszak et al. (2012); Mazur-Marzec et al. (2013); Petersen et al. (2018); Lips and Lips (2017)). Thus, new model approaches
are upcoming, including fuzzy- (e.g., Lilover and Laanemets, 2008; Laanemets et al., 2006) and/or statistical models (e.g.,
Håkanson, 2009). Given the present situation of limited data availability, however, such interesting approaches remain today
rather somewhat inconclusive. We expect that the open data politic of many institutes and public available data collections (like
ICES, HELCOM) facilitate the development of more reliable models.

## 5 Conclusions

The present study summarises current knowledge on cyanobacteria in the Baltic Sea from the peer-reviewed literature. We
consider both, a 'modellers' and a 'biologists' perspective.

There is a certain consensus between the five considered biogeochemical models, that cyanobacteria have an advantage
over other functional groups under nitrate deplete condition. Also, the models agree in that the growth of cyanobacteria is
generally controlled by the availability of light and phosphate. Other than that, there is a relatively large spread on underlying
assumptions: E.g., two models include salinity constrains on the growth of cyanobacteria and one model considers additionally
oxygen, while all others don't. Particularly uncertain seems the determination of critical nutrient thresholds as well as other
model parameters (such as maximum growth rate or temperature dependencies; cf., Table 3 and Fig.1). This uncertainty is even
more pronounced when it comes to the parameterisation of loss processes, such as grazing and mortality (Fig. 2 and Table 4).

So why are model parameter settings and model formulations so different among the current generation of state-of-the-
art models? A comparison of model assumptions with field and laboratory studies suggests following: modellers generally
aim to keep the model complexity on a low level, which typically results in the explicit representation of only one "average"
cyanobacteria species rather than the explicit representation of all potentially important species. The observations indicate that
the most abundant species in the Baltic, *Nodularia spumigena, Aphanizomenon flosaque* and *Dolichospermum sp.* all of which
benefit from very different environmental conditions. E.g. *Aphanizomenon flosaqua.* seems to be able to utilise upwelled
nutrients (e.g. Lips and Lips, 2008)), while *Nodularia spumigena* requires higher temperatures and seems even negatively
affected by upwelling events (Kanoshina et al., 2003). *Nodularia spumigena*, on the other hand, has a higher tolerance to
phosphorus starvation than *Dolichospermum sp.*. Naturally, the attempt to engrain such differing behaviour into one species is
challenging. The alternative, however, to opt for a more complex model with an explicit representation of several species is
also challenging. Increasing model complexity does generally also increases the amount of poorly known model parameters
considerably and, even so, does not necessarily lead to improvements in model performance and predictive skill - rather the
predictive skill can even worsen due to overfitting (e.g. Friedrichs et al., 2007; Kriest et al., 2010; Ward et al., 2013) and many





more to follow). Thus, increasing the number of species in a model requires careful consideration, a good-enough process understanding and a sufficient amount of reliable observations (that resolve the increase in model complexity).

*Author contributions.* All authors contributed to content and structuring of the manuscript. The lead was taken by Britta Munkes.

*Competing interests.* no competing interests are present

*Acknowledgements.* B. Munkes acknowledges support by Deutsche Forschungsgemeinschaft ("Towards a deeper Understanding of Cyanobacteria Blooms in the Baltic Sea"; LO 1377/3-1). Heiner Dietze and Ulrike Löptien are supported by the Helmholtz Association of German Research Centres (HGF) in the project "Reduced Complexity Models" - grant no. ZT-I-0010). Special thanks also to Prof. Ulrich Sommer, who always supported us with his advice.





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



## Tables

**Table 01.** Coupled biogeochemical ocean models considered in this study. Note that models evolve over time. We refer to the referenced versions below.

| Model | Reference |
|---|---|
| BALTSEM | Savchuck, 2002 |
| CEMBS | Dzierzbicka-Głowacka et al. (2013) |
| ECOSMO | Daewel and Schrum (2013) |
| ERGOM | Neumann et al. (2002) |
| SCOBI | Eilola et al. (2009) |





**Table 02.** Description and units of the ecosystem model parameters considered in this study. The focus is on those parameters that differ among the different functional groups.

| Parameter | Description) | Unit |
|---|---|---|
| $\mu$ | max phytoplankton growth rates | $day^{-1}$ |
| $K_P$ | Half-saturation constant for phosphate | $mmolP/m^3$ |
| $mot$ | Linear phytoplankton mortality | $day^{-1}$ |
| $mot_{quad}$ | Quadratic phytoplankton mortality | $day^{-2}$ |
| $\sigma$ | Max. zooplankton grazing | $day^{-1}$ |
| $a_i$ | food preference per functional type i =1,2,3 (optionally multiplied to $\sigma$ ) | $unitless$ |
| $sinki$ | Sinking rate of phytoplankton | $mday^{-1}$ |





**Table 03.** Key model parameters impacting cyanobacteria growth and sinking rates of phytoplankton put into relation to the respective parameters of other functional groups. All models include three functional phytoplankton groups, including cyanobacteria and diatoms. The third functional group, however, is called "flagellates" in ECOSMO, ERGOM and SCOBI, while BALTSEM refers the third functional group to "summer species" and CEMBS to "small phytoplankton". Note, that the ERGOM-model uses a modified Michealis-Menten formulation and the half-saturation constants are thus not directly comparable to the other models. For models which include temperature dependence (Fig01) the values provided refer to $0°C$ (BALTSEM). Note that, as a peculiarity, the BALTSEM models uses different sinking rates for diatoms in spring and autumn and also the sinking speed depends on temperature and morphology.

| Model | $\mu$ (**cyan.**; diatoms; others) | $K_P$ (**cyan**; diatoms, others) | sinki (**cyan.**; diatoms; others) |
|---|---|---|---|
| BALTSEM | **1.0**; 1.3; 0.9 | **1.5**; 1.5; 3.0 | **0.1**; 0.5/0.1; 0.1 |
| CEMBS | **0.33**; 1.3 1.3 | **0.5**; 0.1; 0.05 | not provided |
| ECOSMO | **1.0**; 1.3; 1.1 | **0.05**; 0.05; 0.05 | non-zero ; 0.0; 0.0 |
| ERGOM | **0.5**; 1.0; 0.7 | **0.5** ; 0.3; 0.15 | **-0.1**; 0.5,0.0 |
| SCOBI | **1.0**; 1.3; 0.9 | **0.05**; 0.1; 0.05 | **0.0**; 0.5; 0.1 |





**Table 04.** Model parameters that determine the termination of cyanobacterial blooms. Comparison among the models and respective parameters of other functional groups. The phytoplankton mortality rates in BALTSEM are temperature dependent. The provided values refer to $0°C$. Also, the BALTSEM model uses different parameter values for diatoms in spring and autumn. The provided values for $\sigma$ for the ECOSMO and the SCOBI-model result from the product of max. zooplankton growth rate and food preference for the respective phytoplankton functional groups. The two values for the ECOSMO-model refer to their two zooplankton groups: macro- and micro zooplankton.

| Model | $\sigma$ (**cyan.**; diatoms; other ) | phyt. mot (**cyan.**; diatoms; other ) | phyt. mot$_{quad}$ (**cyan**; diatoms; other ) |
|---|---|---|---|
| BALTSEM | **0.3**; 1.0/0.0; 0.7 | **0.4**; 0.6/0.3; 0.4 | - |
| CMEBS | **0.9**; 1.95; 2.5 | **0.17**; 0.15; 0.15 | **0.0**; 0.0035; 0.0035 |
| ECOSMO | **0.09/0.09**; 0.25/0.68; 0.7/0.08 | **0.08**; 0.05; 0.08 | - |
| ERGOM | **0.25**; 0.5; 0.5 | **0.02**; 0.02; 0.02 | - |
| SCOBI | **0.03**; 0.09; 0.105 | **0.05**; 0.05; 0.05 | - |





**Table 05.** Maximal growth rate ($\mu$) of the studied cyanobacteria species

| Class | species | $\mu$ | Reference |
|---|---|---|---|
| Cyanobacteria | Aphanizomenon flos-aque | 0.39 | Foy et al. (1976b) |
| Cyanobacteria | Aphanizomenon flos-aque | 0.89 | Gotham and Rhee (1981) |
| Cyanobacteria | Aphanizomenon flos-aque | 0.95-1.34 | Rhee and Lederman (1983) |
| Cyanobacteria | Aphanizomenon flos-aque | 0.99 | Lee and Rhee (1999) |
| Cyanobacteria | Aphanizomenon flos-aque | 0.43 | Sommer (1981) |
| Cyanobacteria | Aphanizomenon flos-aque | 0.89 | Gotham and Rhee (1981) |
| Cyanobacteria | Aphanizomenon flos-aque | 0.28-0.31 | Rakko and Seppäälä (2014) |
| Cyanobacteria | Aphanizomenon flos-aque | 0.25-0.75 | Lee and Rhee (1997) |
| Cyanobacteria | Aphanizomenon flos-aque | 0.18 | Konopka and Brock (1978) |
| Cyanobacteria | Aphanizomenon flos-aque, | 1.2 | Robarts and Zohary (1987) |
| Cyanobacteria | Dolichospermum sp. | 0.51 | Foy et al. (1976b) |
| Cyanobacteria | Dolichospermum sp. | 0.41 | Sommer (1981) |
| Cyanobacteria | Dolichospermum sp. | 0.78 | Reynolds (2006) |
| Cyanobacteria | Dolichospermum sp. | 0.4-1.1 | Tang et al. (1997) |
| Cyanobacteria | Dolichospermum sp. | 0.94-1.27 | Oh et al. (1991) |
| Cyanobacteria | Dolichospermum sp. | 0.93 | Lürling et al. (2013) |
| Cyanobacteria | Dolichospermum sp. | 0.4 | Konopka and Brock (1978) |
| Cyanobacteria | Dolichospermum sp. | 1.25 | Nalewajko and Murphy (2001) |
| Cyanobacteria | Nodularia spumigena | 0.14-0.16 | Rakko and Seppäälä (2014) |
| Cyanobacteria | Nodularia spumigena | 0.2-0.6 | Sommer et al. (2006) |
| Cyanobacteria | Nodularia spumigena | 0.13 | Lehtimäki et al. (1997) |





**Table 06.** Maximal growth rate ($\mu$) of different phytoplankton groups

| Class | species | $\mu$ | Reference |
|---|---|---|---|
| Green algae | Chlamydomonas reinhardtii | 3.3-3.8 | Hoogenhout and Amesz (1965) |
| Green algae | Chlorella luteoviridis | 0.56 | Hoogenhout and Amesz (1965) |
| Green algae | Chlorella strain 221 | 1.84 | Reynolds (2006) |
| Green algae | Eudorina unicocca | 0.62 | Reynolds and Rodgers (1983) |
| Green algae | Scenedesmus sp. | 1.35 | Gotham and Rhee (1981) |
| Green algae | Volvox aureus | 0.46 | Reynolds (2006) |
| Green algae | 8 species | 0.24 -1.38 | Lürling et al. (2013) |
| Dinoflagellates | Ceratium furcoides | 0.29 | Butterwick et al. (2005) |
| Dinoflagellates | Ceratium hirundinella | 0.21 | Reynolds (2006) |
| Diatoms | Asterionella formosa | 1.9-2.4 | Hoogenhout and Amesz (1965) |
| Diatoms | Asterionella formosa | 1.78 | Lund (1949) |
| Diatoms | Asterionella formosa | 1.78 | Lund (1949) |
| Diatoms | Asterionella formosa | 0.67 | Holm and Armstrong (1981) |
| Diatoms | Chaetoceros gracilis | >3.1 | Werner (1977) |
| Diatoms | Cyclotella nana | 1.6 | Fuhs (1972) |
| Diatoms | Cyclotella nana | 3.4 | Hoogenhout and Amesz (1965) |
| Diatoms | Fragilaris crotonensis | 0.90 | Gotham and Rhee (1981) |
| Diatoms | Fragilaris crotonensis | 1.37 | Reynolds (2006) |
| Diatoms | Nitzschia actinastroides | 1.5 | Werner (1977) |
| Diatoms | Skeletonema costatum | 4.3 | Hoogenhout and Amesz (1965) |
| Diatoms | Staphanodiscus hantzschii | 1.7 | Hoogenhout and Amesz (1965) |
| Diatoms | Thalassiosira fluviatilis | 1.6 | Fuhs (1972) |





**Table 07.** Optimal temperature range for growth of different phytoplankton groups

| Class | species | °C | Reference |
|---|---|---|---|
| Cyanobacteria | Aphanizomenon flosaquae | 18-22,5 | Tang et al. (1997) |
| Cyanobacteria | Aphanizomenon sp. | 16-22 | Lehtimäki et al. (1994) |
| Cyanobacteria | Aphanizomenon sp. | 25 | Konopka and Brock (1978) |
| Cyanobacteria | Aphanizomenon gracile | 32.5 | Lürling et al. (2013) |
| Cyanobacteria | Dolichospermum sp. | 18-35 | Tang et al. (1997) |
| Cyanobacteria | Dolichospermum sp. | 25 | Lürling et al. (2013) |
| Cyanobacteria | Dolichospermum sp. | 25 | Konopka and Brock (1978) |
| Cyanobacteria | Dolichospermum sp. | 28-32 | Nalewajko and Murphy (2001) |
| Cyanobacteria | Nodularia sp. | 20-25 | Lehtimäki et al. (1994) |
| Cyanobacteria | Nodularia sp. | 25-30 | Nordin et al. (1980) |
| Cyanobacteria | Nodularia sp. | 25-30 | Lehtimäki et al. (1997) |
| Green algae | Ankistrodesmus falcatus | 32.5 | Lürling et al. (2013) |
| Green algae | Chlamydomonos reinhardtii | 27.5 | Lürling et al. (2013) |
| Green algae | Desmodesmus bicellularis | 35 | Lürling et al. (2013) |
| Green algae | Desmodesmus quadricauda | 32.5 | Lürling et al. (2013) |
| Green algae | Monoraphidium minutum | 27.5 | Lürling et al. (2013) |
| Green algae | Scenedesmus acuminatus | 27.5 | Lürling et al. (2013) |
| Dinoflagellates | Ceratium furcoides | 20 | Butterwick et al. (2005) |
| Diatoms | Chaetoceros socialis | 18 | Eppley (1977) |
| Diatoms | Skeletonema costatum (3 studies) | 16-30 | Eppley (1977) |
| Diatoms | Thalassiosira nordenskiolldii | 11-14 | Eppley (1977) |





**Table 08.** Half-saturation constant for phosphate ($K_p$) for different phytoplankton groups.

Please note that different size classes of algae are indicated by numbers (1: tiny; 2: small; 3: medium, 4: large) and by letter (A: chains; B: single; C: other)

| Phylum | Species | $K_p\ [mmol\,P\,m^{-3}]$ | Reference |
|---|---|---|---|
| Cyanobacteria | Aphanizomenon flosaquae[1,2,A] | 1.13 | Gotham and Rhee (1981) |
| Cyanobacteria | Aphanizomenon flosaquae[1,2,A] | 2.5 | (Degerholm et al., 2006) |
| Cyanobacteria | Aphanizomenon flosaquae[1,2,A] | 1-2. | (Healey at al., 1973) |
| Cyanobacteria | Dolichospermum sp.[1,A] | 1.8-2.5 | Reynolds (2006) |
| Cyanobacteria | Microcystis aeruginosa[1,A] | 1.23 | Holm and Armstrong (1981) |
| Cyanobacteria | Microcystis sp.[1,A] | 2.11 | Gotham and Rhee (1981) |
| Cyanobacteria | Nodularia spumigena[2,A] | 1.0-1.7 | Degerholm et al. (2006) |
| Green algae | Chlorella pyrenoidosa[1,2,B] | 0.68 | Nyholm (1977) |
| Green algae | Scenedesmus sp.[1,2,C] | 0.57/3.4 | Gotham and Rhee (1981) |
| Green algae | Volvox aureus[4,C] | 5.24 | Reynolds (2006) |
| Dinoflagellates | Alexandrium tamarense[3,4,B,C] | 2.6 | Yamamoto et al. (2012) |
| Dinoflagellates | Gymnodinium catenatum[4,A,B] | 3.4 | Yamamoto et al. (2012) |
| Dinoflagellates | Peridinium sp.[2−4,B] | 0.11 | Reynolds (2006) |
| Dinoflagellates | Pyrocystis noctiluca[3,4,B] | 2.1 | Rivkin and Swift (1982) |
| Diatoms | Asterionella formosa[1,C] | 0.70 | Holm and Armstrong (1981) |
| Diatoms | Chaetoceros gracilis[1,A,B] | 0.12 | Werner (1977) |
| Diatoms | Cyclotella nana[1,2,A,C] | 0.58 | Fuhs (1972) |
| Diatoms | Nitzschia actinastroides[1,C] | 0.013 | Werner (1977) |
| Diatoms | Skeletonema costatum[1,2,A] | 0.68 | Tarutani and Yamamoto (1994) |
| Diatoms | Thalassiosira fluviatilis[3,4,B] | 1.72 | Fuhs (1972) |





**Table 09.** Optimal light levels ($W/m^2$) for the studied cyanobacteria species

| Species | $W/m^2$ | grow condition | Reference |
|---|---|---|---|
| all 3 species | 61.32 | best | Śliwińska-Wilczewska et al. (2019) |
| Aphanizomenon flos-aque | 6-10 | best | Lehtimäki et al. (1997) |
| Aphanizomenon flos-aque | >33 | best | Üveges et al. (2012) |
| Dolichospermum sp. | 8.32 | best | Eigemann et al. (2018) |
| Dolichospermum sp. | 3.72 | best | Walsby and Booker (1980) |
| Nodularia spumigena | 25 | best | Nordin et al. (1980) |
| Nodularia spumigena | 5-114 | can grow | Nordin et al. (1980) |
| Nodularia spumigena | 32.85 | best | Eigemann et al. (2018) |
| Nodularia spumigena | 23-34 | best | Lehtimäki et al. (1997) |
| Nodularia spumigena | 63.5 | best | Jodlowska and Latala (2019) |
| Nodularia spumigena | 65.7 | best | Roleda et al. (2008) |





**Table 010.** Sinking/ floating speed of different cyanobacteria

| Class | species | $m \cdot d - 1$ | Reference |
|---|---|---|---|
| Cyanobacteria | Aphanizomenon flos-aque | 3.5-5.2 | Reynolds (2006) |
| Cyanobacteria | Aphanizomenon flos-aque | 22 | Walsby et al. (1995) |
| Cyanobacteria | Aphanizomenon flos-aque | 15.5-51.8 | Adam (1999) |
| Cyanobacteria | Doliochospermum sp. | 3.5-5.2 | Reynolds (2006) |
| Cyanobacteria | Dolichospermum sp. | 0.1-0.3 | Walsby et al. (1995) |
| Cyanobacteria | Nodularia spumigena | 36 | Walsby et al. (1995) |
| Cyanobacteria | Nodularia spumigena | 35-45.7 | Adam (1999) |