# Peer review of "Cyanobacteria Blooms in the Baltic Sea: A Review of Models and Facts"

_Biogeosciences, 2020_

## Short Comment (SC1) · 25 May 2020

To my mind, this manuscript provides a very interesting and thorough coverage of important topic. Although I cannot quite agree with some of the author's statements related to modelling approach, the following notes of mine are by no means a formal review of the manuscript, but merely a correction of some confusion and/or misunderstanding.

1) In BALTSEM model, all the variables and biogeochemical fluxes are expressed in the weight, not molar units. As indicated in Table B2 (Savchuk, 2002), the phosphate half-saturation constants for the cyanobacteria growth and nitrogen fixation rates are 1.5 and 9.0 mg P m-3, that is about 0.05 and 0.3 mmol P m-3, respectively. Please, correct

correspondingly in your Table 03; at p. 11, lines 5-7; and elsewhere if necessary.

2) In BALTSEM model, both the mortality rates and sinking velocities of phytoplankton groups depend not only on the water temperature but are also inversely related to the Liebig minimum function as a measure of unfavorable environmental conditions; for cyanobacteria, accounting also for contribution of nitrogen fixation into their total primary production determined by both DIN uptake and nitrogen fixation. In result, neither mortality rate nor sinking velocity have a fixed value, contrary to what is now stated at p. 15, lines 4-6.

---

## Referee Comment (RC1) · Ute Daewel (Referee) · 10 Jul 2020

Ute Daewel (Referee)

ute.daewel@hzg.de

As the title already indicates, the study collects information about the processes related to cyanobacterial blooms in the Baltic Sea and how these are implemented in currently used marine ecosystem models. In that context the authors address processes related to growth, limitation processes as well as mortality. Overall the authors provide a good overview about the cyanobacteria implementation in marine ecosystem models as well as a thorough and very useful review on the available literature on cyanobacteria. However, by now I cannot unreserved recommend the study for publication, as there are some, partly major, points that need to be addressed first.

First: The authors should have gotten in contact to the model developers prior to sub-

mission to verify the model descriptions and implementations. One correction has already issued by Oleg Savchuk, and being one of the developers myself, I have to add some points as well that were not correctly described for the model I use (corrections listed below). This could have easily been avoided by one short communication beforehand. However, I encourage the authors to do so now and correct the section on models accordingly.

Second: The authors give a very detailed review on the processes and impacts on cyanobacteria, in functional group type ecosystem models, however, the basic principle is simplification of the ecosystem. I would appreciate if the authors could more clearly connect the modelling section and the experimental section. The discussion section should be rewritten in a way were the model parameters and the observational finding are related to each other to actually address the "key differences between model approaches and observational evidence" (p. 23, l. 26f) and, if possible, provide recommendations for model improvements.

Third: The manuscript gives the impression that parameterization and choice of functional groups happen at random or according to the modelers needs. While specific model parameters can indeed differ widely ("Somewhat disconcerting, the respective parameter choices differ substantially from one model to another" p. 24, l. 6f), so does observational evidence on which the parameter choices indeed are based. In most cases the developers actually based their parameters on previous experimental publications or at least have a good reason for their assumptions. As this is usually explained in the corresponding publications, it would be more helpful to refer to these underlying reasons for choosing the parameters and revise the impression given in the ms.

The general biological model structures are explained well, but I would consider a brief foray into the physics. This might be too extensive for a paper focusing solely on biological aspects, but some basic explanation of the most important physical variables could be useful for establishing context. Especially as the relationships between cyanobacteria and physics such as temperature are not only directly related as indicated by figure 1 & 2, but also indirectly through the ecosystem interactions as well as physical processes such as transport and upwelling.

Even though I found the ms well readable, the English needs substantial improvement. There are several occasions with wrong grammar, typos and somewhat odd expressions. Some are listed below, but, not being a native speaker myself, I would suggest the authors to edit the language thoroughly.

specific comments:

p2, l 6: "[. . .] a reduction of loads will have no net effect on the nutrient budget because cyanobacteria will compensate. . ." Can you give a citation?

p2, l 12-16: "Some of the numerous studies on cyanobacteria [. . .] are motivated by concerns to run into low-oxygen conditions [. . .]" Please consider reformulation

p2, l 22: comma

p2,l 30: reference from 2006: can you add a more up to date reference

p3,l 30 remove "so-called" as prognostic is a clearly defined concept

p4,l 7 I doubt that any modeler really does "ad-hoc" choices on parameters, please rephrase also see comment above

p4, l 23ff: "[. . .] cyanobacteria grow more slowly [. . .] and can in most models only thrive when nitrogen is no longer accessible to ordinary phytoplankton [. . .]" The citations contain both modelling and observational studies. Does this assumption relate to models or observations?

p4,l 29-31 Can you give a reference where this was done. The phytoplankton bloom dynamics is generally determined by nutrient availability, which is the obvious reason for the nitrogen depletion in surface waters.

P5, l 6 remove "they"

P5, l15-16 please rephrase, its not clear which model does what.

P5, l20 comma

P9 Section 3: In this sections there are some paragraphs that basically just list the same numbers that are given in the tables. You might want to consider shortening these paragraphs to what is new and necessary and avoiding listing the same numbers in the text and in a table.

P10 l23-24 Grammar, please rephrase the two sentences. (& comma)

P11,L8 "Similar to this, . . ."

P15,L28-29 Please explain this sentence.

P18,L27 Reference?

P19,L5 ". . .,they are not able to. . . "

P22, l14-15 revise sentence

p23, l27: "Sect. 4.2 debates the impact of the oceanic processes to the Baltic Sea, respectively" In respect to what?

p24, l9: "Another potentially problematic assumption [. . .] is the fixed Redfield-ratio [. . .]" Why and how is this problematic? Enhance

Specific corrections for ECOSMO:

Temperature dependency for ECOSMO cyanobacteria: $T\_bg=1/(1+exp(-T))$ with growth multiplied by $T\_bg$

-Zooplankton in ECOSMO does not actively feed on itself, but Macrozooplankton feeds on Microzooplankton

-Salinity constraints in ECOSMO: no growth for S > 11.5

-positive buoyancy : sinking vel=-0.1 m/d

---

## Author Comment (AC1) · 17 Jul 2020

Dear Oleg Savchuk,

thank you very much for your time and effort and your constructive comment. We will revise accordingly. Your contribution is very much appreciated!

Kind Regards, the authors

---

## Author Comment (AC2) · 17 Jul 2020

Reviewer #1:

Dear Ute Daewel,

we thank you for your time and effort and constructive comments. Here is a first reply to your comments:

Ute Daewel: As the title already indicates, the study collects information about the processes related to cyanobacterial blooms in the Baltic Sea and how these are implemented in currently used marine ecosystem models. In that context the authors address processes related to growth, limitation processes as well as mortality. Overall the authors provide a good overview about the cyanobacteria implementation in marine
ecosystem models as well as a thorough and very useful review on the available literature on cyanobacteria. However, by now I cannot unreserved recommend the study for publication, as there are some, partly major, points that need to be addressed first.

First: The authors should have gotten in contact to the model developers prior to submission to verify the model descriptions and implementations. One correction has already issued by Oleg Savchuk, and being one of the developers myself, I have to add some points as well that were not correctly described for the model I use (corrections listed below). This could have easily been avoided by one short communication beforehand. However, I encourage the authors to do so now and correct the section on models accordingly.

The authors: We are grateful for your corrections and will acknowledge them accordingly.

Ute Daewel: Second: The authors give a very detailed review on the processes and impacts on cyanobacteria, in functional group type ecosystem models, however, the basic principle is simplification of the ecosystem. I would appreciate if the authors could more clearly connect the modelling section and the experimental section. The discussion section should be rewritten in a way were the model parameters and the observational finding are related to each other to actually address the "key differences between model approaches and observational evidence" (p. 23, l. 26f) and, if possible, provide recommendations for model improvements.

The authors: Thank you for these suggestions. We will improve the link between the modeling and the experimental section and also revise section 4.1 carefully.

Ute Daewel: Third: The manuscript gives the impression that parameterization and choice of functional groups happen at random or according to the modelers needs. While specific model parameters can indeed differ widely ("Somewhat disconcerting, the respective parameter choices differ substantially from one model to another" p. 24, l. 6f), so does observational evidence on which the parameter choices indeed

are based. In most cases the developers actually based their parameters on previous experimental publications or at least have a good reason for their assumptions. As this is usually explained in the corresponding publications, it would be more helpful to refer to these underlying reasons for choosing the parameters and revise the impression given in thems.

The authors: We apologize in case anyone who has written cyanobacteria code feels offended by our text. Our goal was to establish an understanding of the remaining problems, which are associated with simulating cyanobacteria. Please note that we do not rank one model approach over another and we do not deny that there are good reasons for any of the underlying model assumptions used to date.

The problem we see is a dilemma: On one hand, the models are often developed to improve process understanding - as a means to make sense of observational data, which can be contradicting or inherently "differ widely". On the other hand, a comprehensive process understanding appears to be the pre-requisite for developing a mathematical model. The fact that parameter choices differ substantially from one model to another is an indication that these are still subject to discussion. The aim of this review is to identify such knowledge gaps.

We agree that our choice of wording "somewhat disconcerting" is unnecessarily provocative. We will delete this in the revised version of the manuscript.

Ute Daewel: The general biological model structures are explained well, but I would consider a brief foray into the physics. This might be too extensive for a paper focusing solely on biological aspects, but some basic explanation of the most important physical variables could be useful for establishing context. Especially as the relationships between cyanobacteria and physics such as temperature are not only directly related as indicated by figure 1 & 2, but also indirectly through the ecosystem interactions as well as physical processes such as transport and upwelling.

The authors: Agreed. We will add these aspects to the revised version of the

manuscript.

Ute Daewel: Even though I found the ms well readable, the English needs substantial improvement. There are several occasions with wrong grammar, typos and somewhat odd expressions. Some are listed below, but, not being a native speaker myself, I would suggest the authors to edit the language thoroughly.

The authors: We will follow this suggestion. Being non-native speakers we are especially thankful that you took the time for the specific issues listed below.

Ute Daewel: specific comments: p2, l 6: "[...] a reduction of loads will have no net effect on the nutrient budget because cyanobacteria will compensate..." Can you give a citation? The authors: This is a logical consequence in models which assume a fixed Redfield-ratio. We will clarify this point.

Ute Daewel: p2, l 12-16: "Some of the numerous studies on cyanobacteria [...] are motivated by concerns to run into low-oxygen conditions [...]" Please consider reformulation The authors: We will do so.

Ute Daewel: p2, l 22: comma The authors: Thanks.

Ute Daewel: p2,l 30: reference from 2006: can you add a more up to date reference The authors: Agreed. We will add Shimoda et al., 2016 and Taranu et al., 2012 (see below).

Ute Daewel: p3,l 30 remove "so-called" as prognostic is a clearly defined concept A: Agreed.

Ute Daewel: p4,l 7 I doubt that any modeler really does "ad-hoc" choices on parameters, please rephrase also see comment above The authors: We will do so.

Ute Daewel: p4, l 23ff: "[...] cyanobacteria grow more slowly [...] and can in most models only thrive when nitrogen is no longer accessible to ordinary phytoplankton [...]" The citations contain both modelling and observational studies. Does this assumption

relate to models or observations? The authors: It's rather an assumption in most models, supported by observational evidence. We will clarify this in the revised version of our manuscript.

Ute Daewel: p4,l 29-31 Can you give a reference where this was done. The phytoplankton bloom dynamics is generally determined by nutrient availability, which is the obvious reason for the nitrogen depletion in surface waters. The authors: True – we will add a sentence on bottom-up control.

Ute Daewel: P5, l 6 remove "they" The authors: Thanks.

Ute Daewel: P5, l15-16 please rephrase, its not clear which model does what. The authors: We will make this more clear in the revised version of the manuscript.

Ute Daewel: P5, l20 comma The authors: Thanks

Ute Daewel: P9 Section 3: In this sections there are some paragraphs that basically just list the same numbers that are given in the tables. You might want to consider shortening these paragraphs to what is new and necessary and avoiding listing the same numbers in the text and in a table. The authors: Agreed.

Ute Daewel: P10 l23-24 Grammar, please rephrase the two sentences. (& comma) The authors: We will do so.

Ute Daewel: P11,L8 "Similar to this,..." The authors: Thanks

Ute Daewel: P15,L28-29 Please explain this sentence. The authors: We will do so.

Ute Daewel: P18,L27 Reference? The authors: s. below: Eglite et al., 2019/ Wasmund et al., 2019

Ute Daewel: P19,L5 "...,they are not able to..." The authors: Thanks

Ute Daewel: P22, l14-15 revise sentence The authors: Thanks

Ute Daewel: p23, l27: "Sect. 4.2 debates the impact of the oceanic processes to the

Baltic Sea,respectively" In respect to what? The authors: We will delete ",respectively"

Ute Daewel: p24, l9: "Another potentially problematic assumption [...] is the fixed Redfield-ratio[...]" Why and how is this problematic? Enhance The authors: E.g., page 12, line 21ff refers to the cyanobacteria's storage capacity of DIP which is very difficult to account for when assuming a fixed Redfield-ratio. Not considering this in a model might shrink the ecological niche of cyanobacteria. Further, there is observational evidence that diazotrophs do not obey Redfield (e.g., Larsson et al, 2001). We will clarify this in the revised version of the manuscript.

Ute Daewel: Specific corrections for ECOSMO:

Thanks for pointing this out. We will correct all formulations accordingly.

Temperature dependency for ECOSMO cyanobacteria: T_bg=1/(1+exp(-T)) with growth multiplied by T_bg -Zooplankton in ECOSMO does not actively feed on itself, but Macrozooplankton feeds on Microzooplankton -Salinity constraints in ECOSMO: no growth for S > 11.5 -sinking vel=-0.1 m/

additional References: Shimoda, Y., & Arhonditsis, G. B. (2016). Phytoplankton functional type modelling: running before we can walk? A critical evaluation of the current state of knowledge. Ecological Modelling, 320, 29-43.

Taranu, Z. E., Zurawell, R. W., Pick, F., & Gregory‐Eaves, I. (2012). Predicting cyanobacterial dynamics in the face of global change: the importance of scale and environmental context. Global Change Biology, 18(12), 3477-3490.

Larsson, U., Hajdu, S., Walve, J., & Elmgren, R. (2001). Baltic Sea nitrogen fixation estimated from the summer increase in upper mixed layer total nitrogen. Limnology and Oceanography, 46(4), 811-820.

Eglite, E., Graeve, M., Dutz, J., Wodarg, D., Liskow, I., Schulz-Bull, D., and Loick-Wilde, N.: Metabolism and foraging strategies of mid- latitude mesozooplankton during cyanobacterial blooms as revealed by fatty acids, amino acids, and their stable carbon

isotopes, Ecology and Evolution, 9, 9916-9934, 10.1002/ece3.5533, 2019.

Wasmund, N., Dutz, J., Kremp, A., and Zettler, M. L.: Biological assessment of the Baltic Sea 2018, Leibniz Institute for Baltic Sea Research Warnemünde, 10.12754/msr-2018-0108, 2019.

---

## Referee Comment (RC2) · Justus van Beusekom (Referee) · 10 Sep 2020

Review of Cyanobacteria Blooms in the Baltic: A review of Models and Facts By Britta Munkes et al.

In this paper, the authors discuss Cyanobacteria blooms in the Baltic. These blooms may cause environmental problems including toxic events or large biomass productions leading to coastal anoxia. A correct modeling of such blooms is important to check measures to combat eutrophication. The authors aim to bring together both a modelers view and a biologists view. They compare 5 models that are used for political decision making regarding the approach to model cyano-blooms. Against this background, the factors that determine the cyanobacteria blooms in the real world

are discussed. Based on the comparison of models and field/lab studies, the authors conclude that modelers tend to keep models simple (with good reasons), whereas in the real world cyano-blooms are complex with several species each with their specific requirements responding differently to changes in drivers (like nutrients, light, temperature, grazing). The paper is well written, but the discussion and outlook needs more substance. It does not come as a surprise that modelers tend to (have to) keep models simple and that biologists have an eye on complexity. Given this, I expect a more in depth discussion on the next steps to be taken to overcome this schism. For instance, given the focus on those models that are used for policy purposes, the limits of using these models should be discussed and suggestions should be made how this dilemma can be solved. Could for instance the combination of "simple" biogeochemical models and conceptual models be a solution? And are examples available?

Minor comments: P1L12, delete commas around "….severe"

P2L4. This sentence is not very clear but very important. I would add a sentence, that explains the main Question: How will future, management-induced changes in nutrient loads affect the blooms and how does this interact with expected warming?

P2L11 Phosphorus (not ..ous) P1L19: Sedimentary processes: this needs some explanation. Especially feedbacks between cyanos and P-release should at least be mentioned.

P3L4. Alternatives: So in the end I expect suggestions on how superior alternatives can be developed.

P4L6: prognostic variables: add s

P5L31: Typo in .19°C? (dot 19)

P6L32 PAR instead of RAR

P11L30 PO subscript 4, superscript 3 (you swapped both)

P13, header: You discuss Pmax, but no mention is made of alpha. I suggest to add at least some words to this.

P13L13 forNodularia: space between the words

P14L11: Should this be Table 10?

P15L18: Claessen et al describe (no s)

P18L9 Inline -> In line ? (see also P21L33)

P18 L28: hamatusarexs ->hamatus?

P19L24: exceed -> excert?

P21L31: specie ->species

P21L32: add space after (2002)

P23L27: processes on the Baltic. (delete respectively

P25L33: politic->policy?

P25L14: "A comparison of". Do you refer to the present study?

P25L17ff: an unclear sentence. Maybe delete at the end "of which"?

---

## Author Comment (AC3) · 29 Sep 2020

Dear Justus Van Beusekom,

thank you very much for your interesting and useful comments. We will include all of your comments in the revised version of our manuscript:

Justus Van Beusekom: In this paper, the authors discuss Cyanobacteria blooms in the Baltic. These blooms may cause environmental problems including toxic events or large biomass productions leading to coastal anoxia. A correct modelling of such blooms is important to check measures to combat eutrophication. The authors aim to bring together both a modelers view and a biologists view. They compare 5 models that are used for political decision making regarding the approach to model cyano-blooms. Against this background, the factors that determine the cyanobacteria blooms in the real world are discussed. Based on the comparison of models and field/lab studies, the authors conclude that modelers tend to keep models simple (with good reasons), whereas in the real world cyano-blooms are complex with several species each with their specific requirements responding differently to changes in drivers (like nutrients, light, temperature, grazing).

Justus Van Beusekom: The paper is well written, but the discussion and outlook needs more substance. It does not come as a surprise that modelers tend to (have to) keep models simple and that biologists have an eye on complexity. Given this, I expect a more in-depth discussion on the next steps to be taken to overcome this schism. For instance, given the focus on those models that are used for policy purposes, the limits of using these models should be discussed and suggestions should be made how this dilemma can be solved. Could for instance the combination of "simple" biogeochemical models and conceptual models be a solution? And are examples available?

The authors: We find your suggestions very constructive and will revise discussion and outlook accordingly. Specifically, we aim to assess the possible impact of the formulation of cyanobacteria on future projections and make suggestions on how to proceed to solve the dilemma in our revised manuscript. Combining "simple" biogeochemical models and conceptual models might indeed be helpful and we will consider this line of thoughts.

Minor comments:
The authors: Thank you for your careful and detailed review of this manuscript.

Justus Van Beusekom: P1L12, delete commas around ": : :.severe"
The authors: Thanks

Justus Van Beusekom: P2L4. This sentence is not very clear but very important. I would add a sentence, that explains the main question: How will future, management-induced changes in nutrient loads affect the blooms and how does this interact with expected warming?

The authors: We agree that the revised manuscript will benefit from an elaboration of this aspect. We will revise accordingly.

Justus Van Beusekom: P2L11 Phosphorus (not ..ous) P1L19: Sedimentary processes: this needs some explanation. Especially feedbacks between cyanos and P-release should at least be mentioned.
The authors: Thank you for your suggestion, we will include a paragraph about sediment processes and P-release in our revised manuscript.

Justus Van Beusekom: P3L4. Alternatives: so, in the end I expect suggestions on how superior alternatives can be developed.
The authors: Thank you for your comment, we will take these suggestions into account and will add some suggestions towards the development of alternatives.

Justus Van Beusekom: P4L6: prognostic variables: add s
The authors: Thanks

Justus Van Beusekom: P5L31: Typo in .19_C? (dot 19)
The authors: Thanks

Justus Van Beusekom: P6L32 PAR instead of RAR
The authors: Thanks

Justus Van Beusekom: P11L30 POsubscript4, superscript 3 (you swapped both)
The authors: Thanks

Justus Van Beusekom: P13, header: You discuss Pmax, but no mention is made of alpha. I suggest to add at least some words to this.
The authors: You are right, we will include alpha in our revised manuscript.

Justus Van Beusekom: P13L13 for Nodularia: space between the words
The authors: Thanks

Justus Van Beusekom: P14L11: Should this be Table 10?
The authors: It's great that you noticed this. We will correct the number.

Justus Van Beusekom: P15L18: Claessen et al describe (no s)
The authors: Thanks

Justus Van Beusekom: P18L9 Inline -> In line ? (see also P21L33)
The authors: Thank you for this note. We will correct all wrong "in line" in the revised version.

Justus Van Beusekom: P18 L28: hamatusarexs ->hamatus?
The authors: Thank you for this note. You are correct with this species name.

Justus Van Beusekom: P19L24: exceed -> excert?
The authors: Thanks

Justus Van Beusekom: P21L31: specie ->species

The authors: Thanks

Justus Van Beusekom: P21L32: add space after (2002)
The authors: Thanks

Justus Van Beusekom: P23L27: processes on the Baltic. (delete respectively
The authors: Thanks, we will do so.

Justus Van Beusekom: P25L33: politic->policy?
The authors: That is right, thank you.

Justus Van Beusekom: P25L14: "A comparison of". Do you refer to the present study?
The authors: Yes, we refer to models and field experiments studied in this manuscript. We will change the sentence, so it will be easier to understand.

Justus Van Beusekom: P25L17ff: an unclear sentence. Maybe delete at the end "of which"?
The authors: That is right, thank you.

---

## Author Response (AR1)

Kiel, 6th of Nov., 2020
Dr. Britta Munkes Dr. Ulrike Löptien, Dr. Heiner
Dietze
Helmholtz Centre for Ocean Research
Geomar
Düsterbrooker Weg 20
D - 24105 Kiel
bmunkes@geomar.de
uloeptien@geomar.de
hdietze@geomar.de

Dear Perran Cook,

we herewith submit a substantially revised version of our manuscript titled 'Cyanobacteria Blooms in the Baltic Sea: A Review of Models and Facts'. All reviewers and Oleg Savchuk provided very helpful and constructive comments. In addition to their suggestions we carefully scanned the text for language issues. We hope that the improved manuscript now meets the standard for publication in Biogeosciences.

Point-by-point responses to the reviewer comments are listed below. We find that in all cases the reviewers work has clearly improved our manuscript. Please note that line numbers refer to the manuscript version that includes marked changes bold.

We are grateful for your and the reviewer's work.

Yours Sincerely,
the authors

**Point-to-point responses to the Reviewers:**

Reviewer #1, Ute Daewel:
Ute Daewel: As the title already indicates, the study collects information about the processes related to cyanobacterial blooms in the Baltic Sea and how these are implemented in currently used marine ecosystem models. In that context the authors address processes related to growth, limitation processes as well as mortality. Overall the authors provide a good overview about the cyanobacteria implementation in marine ecosystem models as well as a thorough and very useful review on the available literature on cyanobacteria. However, by now I cannot unreserved recommend the study for publication, as there are some, partly major, points that need to be addressed first.

First: The authors should have gotten in contact to the model developers prior to submission to verify the model descriptions and implementations. One correction has already issued by Oleg Savchuk, and being one of the developers myself, I have to add some points as well that were not correctly described for the model I use (corrections listed below). This could have easily been avoided by one short communication beforehand. However, I encourage the authors to do so now and correct the section on models accordingly.
*The authors: Now, we have been in contact with the model developers and made some corrections accordingly:*

Ute Daewel: Second: The authors give a very detailed review on the processes and impacts on cyanobacteria, in functional group type ecosystem models, however, the basic principle is simplification of the ecosystem. I would appreciate if the authors could more clearly connect the modelling section and the experimental section.

*The authors: Throughout the document, we made changes in order to improve the link between observational biological evidence and the modelling sections. In each chapter about the biology of cyanobacteria there is, now, also a comparison of how this biological aspect is applied in the studied models. More specifically respective additions have been added at:*
*Chapter 3.1 Growth: P 9, L 10.*
*Chapter 3.1.2 Temperature dependence: P 10, L 20-22.*
*Chapter 3.1.3 Nutrient demands: P 11, L 15-19 & P 12, L 31-32 & P 13, L 34-25 .*
*Chapter 3.1.4 Light limitation: P 13, L 21-23.*
*Chapter 3.1.5 Buoyancy: P 14, L14-18..*
*Chapter 3.2 Loss Terms: P 16, L 3-7.*
*Chapter 3.2.4 Grazing on cyanobacteria: P 19, L 6-7 & P 19, L 27-30.*
*Chapter 3.3.1. Salinity constrains: P 22, L 31 & 23, L 1-2 & P 23, L 3-5.*
*Chapter 3.3.2 Cyanobacteria akinetes: P 24, L 1-3.*

Ute Daewel: The discussion section should be rewritten in a way were the model parameters and the observational finding are related to each other to actually address the "key differences between model approaches and observational evidence" (p. 23, l. 26f) and, if possible, provide recommendations for model improvements.
*The authors: We addressed suggestions of the reviewer and discuss in more detail the biological findings in comparison with the model parameters. More specifically we added:*
*Chapter 4.1 Biogeochemical processes: P 26, L 24-32 & P 27, L 1-5.*
*Chapter 4.2 Abiotic oceanic processes: P 26, L 7-15 & P 26, L 25-29.*
*Chapter 4.4 Future Perspectives: P 27, L 24-28 & P 28, L 12-15.*
*Chapter 5 Conclusion: P 29, L 11-15.*

Ute Daewel: Third: The manuscript gives the impression that parameterization and choice of functional groups happen at random or according to the modelers needs. While specific model parameters can indeed differ widely ("Somewhat disconcerting, the respective parameter choices differ substantially from one model to another" p. 24, l. 6f), so does observational evidence on which the parameter choices indeed are based. In most cases the developers actually based their parameters on previous experimental publications or at least have a good reason for their assumptions. As this is usually explained in the corresponding publications, it would be more helpful to refer to these underlying reasons for choosing the parameters and revise the impression given in them.

*The authors: We apologize in case anyone who has written cyanobacteria code feels offended by our text. Our goal was to establish an understanding of the remaining problems, which are associated with simulating cyanobacteria. Please note that we do not rank one model approach over another and we do not deny that there are good reasons for any of the underlying model assumptions used to date.*
*Sorry. The phrase "somewhat disconcerting" was obviously offending. We deleted it and reformulated the section:*
*Chapter 4.1 Biogeochemical processes: P 25, L 19-24.*

Ute Daewel: The general biological model structures are explained well, but I would consider a brief foray into the physics. This might be too extensive for a paper focusing solely on biological aspects, but some basic explanation of the most important physical

variables could be useful for establishing context. Especially as the relationships between cyanobacteria and physics such as temperature are not only directly related as indicated by figure 1 & 2, but also indirectly through the ecosystem interactions as well as physical processes such as transport and upwelling.
*The authors: Agreed. We add these aspects to the revised version of the manuscript.*
*Chapter 4.2 Abiotic oceanic processes: P 26, L 7-15.*

Ute Daewel: Even though I found the ms well readable, the English needs substantial improvement. There are several occasions with wrong grammar, typos and somewhat odd expressions. Some are listed below, but, not being a native speaker myself, I would suggest the authors to edit the language thoroughly.
*The authors: We followed this suggestion. We have checked the manuscript again for spelling errors and odd expression.*

Ute Daewel: specific comments: p2, l 6: "[...] a reduction of loads will have no net effect on the nutrient budget because cyanobacteria will compensate..." Can you give a citation?
*The authors: This is a logical consequence in models which assume a fixed Redfield-ratio. The missing reference: P 2, L 6: Molot et al. (2014).*

Ute Daewel: p2, l 12-16: "Some of the numerous studies on cyanobacteria [...] are motivated by concerns to run into low-oxygen conditions [...]" Please consider reformulation
*The authors: The sentence was reformulated for clarification.*
*P 2, L 10-12.*

Ute Daewel: p2, l 22: comma
*The authors: Thanks.*

Ute Daewel: p2,l 30: reference from 2006: can you add a more up to date reference
*The authors: Agreed. We will add Shimoda et al., 2016 and Taranu et al., 2012 (see below).*
*P 2, L 35.*

Ute Daewel: p3,l 30 remove "so-called" as prognostic is a clearly defined concept A:
*The authors: Agreed.*
*P 3, L 33-2.*

Ute Daewel: p4,l 7 I doubt that any modeler really does "ad-hoc" choices on parameters, please rephrase also see comment above
*The authors: We rephrased the sentence.*
*P 4, L 10-11.*

Ute Daewel: p4, l 23ff: "[...] cyanobacteria grow more slowly [...] and can in most models only thrive when nitrogen is no longer accessible to ordinary phytoplankton [...]" The citations contain both modelling and observational studies. Does this assumption relate to models or observations?
*The authors: It is an assumption implicit to most models, supported by observational evidence. We changed the references in the revised version of our manuscript to make the context clearer.*
*P 4, L 26-29.*

Ute Daewel: p4,l 29-31 Can you give a reference where this was done. The phytoplankton bloom dynamics is generally determined by nutrient availability, which is the obvious reason for the nitrogen depletion in surface waters.
*The authors: True – we added a sentence on bottom-up control.*
*P 4, L 30-32 & P 5, L 33-5.*

Ute Daewel: P5, l 6 remove "they"
*The authors: Thanks.*

Ute Daewel: P5, l15-16 please rephrase, its not clear which model does what.
*The authors: We changed the sentence in the revised version of the manuscript to clarify the differences between the models.*
*P 5, L 19-23.*

Ute Daewel: P5, l20 comma
*The authors: Thanks.*

Ute Daewel: P9 Section 3: In this sections there are some paragraphs that basically just list the same numbers that are given in the tables. You might want to consider shortening these paragraphs to what is new and necessary and avoiding listing the same numbers in the text and in a table.
*The authors: Agreed. We have shortened the sentences, removed superfluous information, and listed the content in the form of bullet points. Please find our changes at:*
*P 10, L 10-16.*
*P 11, L 1-7.*
*P 14, L 34-4.*
*P 15, L 9-17.*
*P 23, L 13-27.*
*P 24, L 18- 33.*

Ute Daewel: P10 l23-24 Grammar, please rephrase the two sentences. (& comma)
*The authors: We rephrased and shortened the sentences.*
*P 11, L 1-2.*

Ute Daewel: P11,L8 "Similar to this,…"
*The authors: Thanks*

Ute Daewel: P15,L28-29 Please explain this sentence.
*The authors: We rephrased the sentence to improve the explanation of the link between PCD and biogeochemical cycling.*
*P 16, L 27-29.*

Ute Daewel: P18,L27 Reference?
*The authors: see reference Eglite et al., 2019/ Wasmund et al., 2019 added to:*
*P 20, L 6.*

Ute Daewel: P19,L5 "…,they are not able to…"
*The authors: Thanks*
*P 20, L 30.*

Ute Daewel: P22, l14-15 revise sentence
*The authors: Thanks. Please see:*
*P 23, L 25-27.*

Ute Daewel: p23, l27: "Sect. 4.2 debates the impact of the oceanic processes to the Baltic Sea, respectively" In respect to what?
*The authors: We deleted ",respectively"*
*P 25, L 12-13.*

Ute Daewel: p24, l9: "Another potentially problematic assumption […] is the fixed Redfield-ratio[…]" Why and how is this problematic? Enhance

*The authors: e.g., page 13, L 33-7 refers to the cyanobacteria's storage capacity of DIP which is very difficult to account for when assuming a fixed Redfield-ratio. Not considering this in a model might shrink the ecological niche of cyanobacteria. Further, there is observational evidence that diazotrophs do not obey Redfield (e.g., Larsson et al, 2001). We rephrased the sentence and made the context clear in the revised version of the manuscript.*
*P 25, L 26-32.*

Ute Daewel:   Specific corrections for ECOSMO:
*The authors: Thanks for pointing this out. We changed the model description accordingly.*

Ute Daewel:  Temperature dependency for ECOSMO cyanobacteria: T_bg=1/(1+exp(-T)) with growth multiplied by T_
*The authors: Changed in Fig. 1 and P 6, L 3.*

Ute Daewel: bg-Zooplankton in ECOSMO does not actively feed on itself, but Macrozooplankton feeds on Microzooplankton
*The authors: Sorry. Changed (P  9, L 7).*

Ute Daewel: Salinity constraints in ECOSMO: no growth for S > 11.
*The authors: We fixed that (P 5, L 21).*

Ute Daewel: positive buoyancy : sinking vel=-0.1 m/d
*The authors: Changed in Tab.3.*

Reviewer #2, Justus Van Beusekom:

Justus Van Beusekom: In this paper, the authors discuss Cyanobacteria blooms in the Baltic. These blooms may cause environmental problems including toxic events or large biomass productions leading to coastal anoxia. A correct modelling of such blooms is important to check measures to combat eutrophication. The authors aim to bring together both a modelers view and a biologists view. They compare 5 models that are used for political decision making regarding the approach to model cyano-blooms. Against this background, the factors that determine the cyanobacteria blooms in the real world are discussed. Based on the comparison of models and field/lab studies, the authors conclude that modelers tend to keep models simple (with good reasons), whereas in the real world cyano-blooms are complex with several species each with their specific requirements responding differently to changes in drivers (like nutrients, light, temperature, grazing).
The paper is well written, but the discussion and outlook needs more substance. It does not come as a surprise that modelers tend to (have to) keep models simple and that biologists have an eye on complexity. Given this, I expect a more in-depth discussion on the next steps to be taken to overcome this schism. For instance, given the focus on those models that are used for policy purposes, the limits of using these models should be discussed and suggestions should be made how this dilemma can be solved. Could for instance the combination of "simple" biogeochemical models and conceptual models be a solution? And are examples available?

*The authors: We find your suggestions very constructive and changed the discussion and outlook accordingly. Specifically, we included an additional chapter: '4.4. Future Perspectives' to assess the possible impact of the formulation of cyanobacteria on future projections and make (now) suggestions on how to proceed to solve the dilemma in our revised manuscript.*
*P 27, L 24-33 and P 28.*

Minor comments:
*Justus Van Beusekom: P1L12, delete commas around ": : :.severe"*
*The authors: Thanks*
*P 1, L 11.*

Justus Van Beusekom: P2L4. This sentence is not very clear but very important. I would add a sentence, that explains the main question: How will future, management-induced changes in nutrient loads affect the blooms and how does this interact with expected warming?
*The authors: We addressed this question in chapter '4.4. Future Perspectives'*
*P 27, L 24-33 and P 28.*
*P 2, L 1-3*

Justus Van Beusekom: P2L11 Phosphorus (not ..ous) P1L19: Sedimentary processes: this needs some explanation. Especially feedbacks between cyanos and P-release should at least be mentioned.
*The authors: Thank you for your suggestion, we included a paragraph about sediment processes and P-release in our revised manuscript.*
*P 2, L 18-22.*

Justus Van Beusekom: P3L4. Alternatives: so, in the end I expect suggestions on how superior alternatives can be developed.
*The authors: Following your suggestions we added the chapter '4.4. Future Perspectives'*
*P 27, L 24-33 and P 28.*
*P 29, L 20-24.*

Justus Van Beusekom: P4L6: prognostic variables: add s
*The authors: Thanks.*

Justus Van Beusekom: P5L31: Typo in .19_C? (dot 19)
*The authors: Thanks.*

Justus Van Beusekom: P6L32 PAR instead of RAR
*The authors: Thanks.*

Justus Van Beusekom: P11L30 POsubscript4, superscript 3 (you swapped both)
*The authors: Thanks.*

Justus Van Beusekom: P13, header: You discuss Pmax, but no mention is made of alpha. I suggest to add at least some words to this.
*The authors: We rewrote the paragraph and have included some explanations about alpha in our revised manuscript.*
*P 13, L 10-14.*
*P 13, L 26-31.*

Justus Van Beusekom: P13L13 for Nodularia: space between the words
*The authors: Thanks.*

Justus Van Beusekom: P14L11: Should this be Table 10?
*The authors: It's great that you noticed this. We corrected the reference to the table.*
*P 15, L 7.*

Justus Van Beusekom: P15L18: Claessen et al describe (no s)
*The authors: Thanks.*

Justus Van Beusekom: P18L9 Inline -> In line ? (see also P21L33)
*The authors: Thank you for this note. We corrected all wrong "in line" in the revised version.*
*P 19, L 21.*
*P 23, L 15.*
*P 23, L 31.*

Justus Van Beusekom: P18 L28: hamatusarexs ->hamatus?
*The authors: Thank you for this note. You are correct with this species name. We changed the species name.*

*P 20, L 6.*

Justus Van Beusekom: P19L24: exceed -> excert?
The authors: Thanks.

Justus Van Beusekom: P21L31: specie ->species
*The authors: Thanks.*

Justus Van Beusekom: P21L32: add space after (2002)
*The authors: Thanks.*

Justus Van Beusekom: P23L27: processes on the Baltic. (delete respectively
*The authors: Thanks, we deleted the respectively.*

Justus Van Beusekom: P25L33: politic->policy?
*The authors: That is right, thank you.*

Justus Van Beusekom: P25L14: "A comparison of". Do you refer to the present study?
*The authors: Yes, we refer to models and field experiments studied in this manuscript. We changed the paragraph, so it is easier to understand.*
P 29, L 16-19.

Justus Van Beusekom: P25L17ff: an unclear sentence. Maybe delete at the end "of which"?
*The authors: That is right, thank you. We changed and shortened the whole chapter and deleted the sentence.*

Summary
9/11/20 6:54:47 PM

Differences exist between documents.

**New Document:**
cyanoreviewCorrectionBM7
60 pages (1.21 MB)
9/11/20 6:54:05 PM
Used to display results.

**Old Document:**
bg-2020-151-manuscript-munkes-1
54 pages (1.60 MB)
9/11/20 6:53:59 PM

Get started: first change is on page 1.

No pages were deleted

**How to read this report**

Highlight indicates a change.
Deleted indicates deleted content.
▲ indicates pages were changed.
↔ indicates pages were moved.

[revised manuscript text omitted]

---

## Author Response (AR2)

List of all relevant changes made in the manuscript bg-2020-151-Munkes:

(1) We went carefully through the manuscript and revised the language.

(2) We performed the suggested corrections in the model formulations (Sect. 2) and revised in this context also Fig.1.

(3) We rephrased potentially offensive expressions in the modelling part and extended the explanation of the dilemma of modellers (e.g., Sect.4.4, Conclusions).

(4) We shortened the biological part (i.e. description of the tables) considerably and enhanced the readability by introducing bullet points, whenever the different species are considered (Sect.3).

(5) We refer now systematically to the respective model assumptions when summarizing the current biological knowledge (Sect.3)

(6) We extended the Section 4.2. (Abiotic Processes) according the reviewer's suggestions and discuss the interactions between cyanobacteria and the physical processes in more detail now.

(7) We introduced a new subsection "Future perspectives" to the Discussion and elaborate on potential future changes of cyanobacteria blooms, including the potential impact of management actions.

(8) The new subsection "Future perspectives" (Sect.4.4) additionally suggests now measures to enhance the reliability of projections.

(9) We rewrote the Conclusions.

(10) We added some extra explanation according to the reviewer's suggestions:
- We explained the interactions between low-oxygen conditions and cyanobacteria processes (Sect.1)
- We included a paragraph on sediment processes and phosphate feedbacks (Sect.1).
- We inserted some extra explanations about photosynthetic processes, including alpha (Sect.3).
- We extended the explanation of the connection between programmed cell death and biochemical cycling (Sect.3).
- We explain now why assuming a fixed Redfield-ratio in models might be problematic (Discussion).

(11)  We inserted additional references to the revised manuscript.

Kiel, 6th of Nov., 2020
Dr. Britta Munkes Dr. Ulrike Löptien, Dr. Heiner Dietze
Helmholtz Centre for Ocean Research Geomar
Düsterbrooker Weg 20
D - 24105 Kiel
bmunkes@geomar.de
uloeptien@geomar.de
hdietze@geomar.de

Dear Perran Cook,

we herewith submit a substantially revised version of our manuscript titled 'Cyanobacteria Blooms in the Baltic Sea: A Review of Models and Facts'.

As you summarized, and of key importance towards an improved version of the manuscript was to put more effort into reconciling the reductionists approach of modellers and the more complex view of ecologists. Further, you raised the pressing question where increased complexity is needed to better reproduce key observations. Following these recommendations, we added more detail to an intertwined discussion of the modellers approach and the ecologists. However, rather than reconciling the modellers approach and the ecologists view it became even more apparent how big of a challenge is ahead of us on our way towards reliable model forecasts. We hope that, to this end, our review will be a first step in that it provides a comprehensive summary of the state-of-the-art.

Please note that we have made a serious effort to improve the accessibility of the information collated in this review (a total of more than 1000 changes following the advice of the very constructive reviews). Because we rewrote so much we invented the following system in our point by point responses: New, revised text is marked in font colour green. Blue denotes the original text into which we embedded our (green) changes.

Please note that the revised version of our manuscript with tracked-changes is best viewed with Adobe. We apologize for any inconvenience!

We hope that the improved manuscript now meets the standard for publication in Biogeosciences.

In any case we are grateful for your and the reviewer's work.

Yours Sincerely,
the authors

**Point-to-point responses to the Reviewers:**

**Reviewer #1, Ute Daewel:**

Dear Ute Daewel,
thank you very much for your time and effort. Based on your (and the other reviewer's help) we rewrote so much we invented the following system in our point-by-point responses: New, revised text is marked in font colour green. Blue denotes the original text into which we embedded our (green) changes.

Kind regards,
the authors

**Ute Daewel:** As the title already indicates, the study collects information about the processes related to cyanobacterial blooms in the Baltic Sea and how these are implemented in currently used marine ecosystem models. In that context the authors address processes related to growth, limitation processes as well as mortality. Overall the authors provide a good overview about the cyanobacteria implementation in marine ecosystem models as well as a thorough and very useful review on the available literature on cyanobacteria. However, by now I cannot unreserved recommend the study for publication, as there are some, partly major, points that need to be addressed first.

First: The authors should have gotten in contact to the model developers prior to submission to verify the model descriptions and implementations. One correction has already issued by Oleg Savchuk, and being one of the developers myself, I have to add some points as well that were not correctly described for the model I use (corrections listed below). This could have easily been avoided by one short communication beforehand. However, I encourage the authors to do so now and correct the section on models accordingly.

*The authors: Thank you for your offer to contact you and your help! We have been in contact with the model developers and made corrections accordingly. For BALTSEM we corrected model parameters accordingly to comments of Oleg Savchuk (see response to the comment by Oleg). For ECOSMO we changed the model description accordingly to Ute Daewel (see below):*
**Ute Daewel**: Temperature dependency for ECOSMO cyanobacteria: $T\_bg=1/(1+\exp(-T))$ with growth multiplied by $T\_$
*The authors: Changed in Fig. 1 and text. Page 6, line 3-6 now reads:*
*"The ERGOM-model stalls all cyanobacteria growth below 12°C and sets maximum growth at temperatures exceeding 19°C. In contrast, the assumed increase in growth with temperature is rather gradual in the CEMBS model. The ECOSMO-model includes a comparably weak temperature dependence for cyanobacteria growth (pers. comm. Ute Daewel)."*

**Ute Daewel**: bg-Zooplankton in ECOSMO does not actively feed on itself, but Macrozooplankton feeds on Microzooplankton
*The authors: Changed. Page 9, line 5ff now reads:*
*"........ the ECOSMO-model differentiates two zooplankton groups (micro- and macro zooplankton) and assumes that micro-zooplankton feeds on phytoplankton and detritus, while macro zooplankton feeds additionally on micro zooplankton."*

**Ute Daewel**: Salinity constraints in ECOSMO:
*The authors: After talking to Ute and in accordance with the respective paper we changed the sentence. Page 5, line 23f now reads:*
*"The SCOBI and ECOSMO-models include an additional switch which shuts down cyanobacterial growth at salinities above 10 and 11.5 PSU, respectively."*

**Ute Daewel**: positive buoyancy : sinking vel=-0.1 m/d
*The authors*: *Changed in Tab.3, page 54.*

**Ute Daewel**: Second: The authors give a very detailed review on the processes and impacts on cyanobacteria, in functional group type ecosystem models, however, the basic principle is simplification of the ecosystem. I would appreciate if the authors could more clearly connect the modelling section and the experimental section.

*The authors:* *Throughout the document, we made changes in order to improve the link between observational biological evidence and the modelling sections. In each chapter about the biology of cyanobacteria there is, now, also a comparison of how this biological aspect is implemented in the studied models. More specifically respective additions have been added (in green):*

*Chapter 3.1.2 Temperature dependence. Page 10, line 21-25 now reads:*
*"In terms of temperature dependency, the model formulations differ widely and some models (e.g., SCOBI and BALTSEM) assume that cyanobacteria require higher temperatures than ordinary phytoplankton for optimal growth (Fig. 1). The respective model assumptions are roughly in agreement with experimental work: while cyanobacteria species typically have optimal growth at higher temperatures than dinoflagellates or diatoms (Paerl et al., 2011), there are only small differences between cyanobacteria and green algae (Lürling et al., 2013). However, there is considerable variation between species."*

*Chapter 3.1.4 Light limitation. Page 13, line 12-26 now reads:*
*"Cyanobacterial and algal photosynthesis rates are significantly influenced i.a. by the combination of light intensity and temperature (Butterwick et al., 2005)......... This basic concept that algal growth is influenced by light and temperature is captured by all the models considered here - although it comes in different flavours: ECOSMO and CEMBS assume that the light requirement for cyanobacteria is higher than for other functional groups, while the other models do not distinguish between functional groups here (c.f., Sect..2.2 )."*

*Chapter 3.1.5 Buoyancy. Page 14, line 17-21 now reads:*
*"Many cyanobacteria possess buoyancy regulation mechanisms, which enables them to actively control their position in the water column (Visser et al., 2016). In accordance, the ERGOM and ECOSMO model assume that cyanobacteria are buoyant, while the SCOBI model applies a lower sinking speed for cyanobacteria than for other phytoplankton. All respective velocities are in the order of centimetres per day and do not change with light intensity or nutrient availability (Tab. 3)."*

*Chapter 3.2 Loss Terms. Page 15, line 30-32 & page 16, line 1-7 now reads:*
*"There are a number of lethal threats for cyanobacteria cells: cell death can be caused by necrosis through adverse environmental conditions, such as insufficient light, nutrients or temperature, or by a programmed cell death (PCD) (Franklin, 2013). Cells can be infected by fungi, undergo viral lysis (Munn, 2011) or can be grazed by a diverse selection of zooplankton (Franklin, 2013). The following subsections list the various causes for cyanobacteria loss. Note that the biogeochemical models investigated in this study differentiate only grazing from other causes of cell losses. For all non-grazing related losses, the models generally assume a fixed loss rate which depends either linearly or quadratic on abundance. An exception is the BALTSEM model, where the mortality of phytoplankton depends on water temperature and is also inversely related to the Liebig minimum function (as a measure of unfavourable environmental conditions; c.f., Sect.2.3)."*

*Chapter 3.2.4 Grazing on cyanobacteria. Page 19, line 5-8 & line 26-31 now reads:*
*"Several studies suggested that there is hardly any grazing on cyanobacteria due to their toxicity, bioactive compounds that hamper digestion, bad taste, poor content of lipids, large filamentous size, and low food quality (Carey et al., 2012; Daewel and Schrum, 2013; Ger et al., 2016). All considered models agree in that cyanobacteria are grazed less than other phytoplankton, while the precise proportions vary between the models (Tab.4).*
*...........This rather low grazing pressure on cyanobacteria has a huge ecosystem impact, as, consequently, during cyanobacteria blooms a lower proportion of the primary production is consumed by larger grazers and therefore not transferred to higher trophic levels. These findings are represented in the models by assuming that grazing on diatoms or other phytoplankton is 2-4 times higher than grazing on cyanobacteria. There is, however, no*

*consensus yet on the exact formulation of zooplankton grazing in the current model generation and grazers are represented by a single (BALTSEM, ERGOM, SCOBI and CEMBS) or two (ECOSMO) functional groups."*

*Chapter 3.3.1. Salinity constrains. Page 22, line 27f & 31f & page 23, line 1-6 now reads:*
*"The Baltic Sea features a wide range of salinities, ranging from 15-25 PSU in the north western part of the Baltic to 2-3 PSU in the Bothnian Bay........ By this means- salinity has the potential to control the occurrence of cyanobacteria species. Salinity thresholds are set in SCOBI and ECOSMO where growth is not permitted above 10 and 11.5 PSU, respectively. The other models do not include salinity constrains on simulated cyanobacteria. Field observations show that in most parts of the Baltic Sea large cyanobacteria blooms occur during summer, except in the relatively saline waters in the Kattegat and the Belt Sea. Thus, e.g., Rakko and Seppäälä (2014) conclude, in line with the SCOBI and ECOSMO models, that high salinities seem to restrict the growth of Baltic Sea cyanobacteria and estimate a threshold around 10 PSU."*

**Ute Daewel**: The discussion section should be rewritten in a way were the model parameters and the observational finding are related to each other to actually address the "key differences between model approaches and observational evidence" (p. 23, l. 26f) and, if possible, provide recommendations for model improvements.

***The authors***: *We addressed suggestions of the reviewer and discuss in more detail the biological findings in comparison with the model parameters in the Sect. 4.1 (Biogeochemical Processes). The newly introduced Subsection 4.4. (Future Perspectives) contains several suggestions for measures to improve the reliability of projections. Also, we rewrote the "Conclusions" to highlight potential steps forward.*
*More specifically we added:*

*Chapter 4.1 Biogeochemical processes. Page 25, line 23-31 & page 26, line 1-4 now reads:*

[revised manuscript text omitted]

**Ute Daewel**: Third: The manuscript gives the impression that parameterization and choice of functional groups happen at random or according to the modelers needs. While specific model parameters can indeed differ widely ("Somewhat disconcerting, the respective parameter choices differ substantially from one model to another" p. 24, l. 6f), so does observational evidence on which the parameter choices indeed are based. In most

cases the developers actually based their parameters on previous experimental publications or at least have a good reason for their assumptions. As this is usually explained in the corresponding publications, it would be more helpful to refer to these underlying reasons for choosing the parameters and revise the impression given in them.

*The authors: We apologize in case anyone who has written cyanobacteria code feels offended by our text. Our goal was to establish an understanding of the remaining problems, which are associated with simulating cyanobacteria. Please note that we do not rank one model approach over another and we do not deny that there are good reasons for any of the underlying model assumptions used to date. We reformulated the "Conclusions", which now explains the difficult situation of modelers and suggests measures to overcome these. Also, we deleted the phrase "somewhat disconcerting" as it seemed offending:*

*Chapter 4.1 Biogeochemical processes. Page 25, line 22f now reads:*
*"The respective parameter choices differ substantially from one model to another, depending on the focus of the respective modeller (Table 3 and 4; Fig. 1 and 2)."*

**Ute Daewel:** The general biological model structures are explained well, but I would consider a brief foray into the physics. This might be too extensive for a paper focusing solely on biological aspects, but some basic explanation of the most important physical variables could be useful for establishing context. Especially as the relationships between cyanobacteria and physics such as temperature are not only directly related as indicated by figure 1 & 2, but also indirectly through the ecosystem interactions as well as physical processes such as transport and upwelling.

*The authors: Agreed. We added an explanation of physical drivers to the revised version of the manuscript and refer now to the respective ocean models for all five biogeochemical models:*

*Chapter 4.2 Abiotic oceanic processes. Page 26, line 6-23 now reads:*
*"The occurrence of cyanobacteria is controlled by several abiotic factors, such as nutrient availability, light, temperature, salinity, mixed layer depth, currents and upwelling. This control is both direct and indirect. E.g. temperature directly affects the speed of enzymatic reactions and as such photosynthesis rates and respiration. Temperature will also directly affect organism by changing the oxygen concentration in seawater and its viscosity. Indirect controls include feedback loops where temperature changes the interactions and competition in the feedback (e.g. by promoting competitors or predators). Similar effects exists for surface mixed layer dynamics which modulates the average light levels experienced by phytoplankton cells dispersed in the surface layer. A deepening may, e.g. promote buoyant species which manage to stay at the sun-lit surface while competitors and grazers are mixed downwards into dark ocean layers. Further complexity comes from abiotic nutrient transport which is determined by mixing processes and advection.*
*Given the prominent controls of abiotic processes on pelagic ecosystem dynamics it is desirable to ensure utmost realism in reproducing abiotic drivers. CEMBS, ECOSMO, ERGOM and SCOBI attempt this by coupling to full general ocean circulation models (GCMs) which explicitly calculate three dimensional current fields in response to wind and buoyancy fluxes prescribed at the surface. The underlying GCMs (POP, HAMSOM, MOM and RCO, respectively) are conceptually very similar. In terms of spatial resolution, however, they span a wide range (1 to 5nm horizontally and 2 to 5m vertically). This suggests that their capability of reproducing small-scale processes such as local upwelling events will also span a wide range because, those models incapable of explicitly resolving the flow fields, need to parametrize their effect by mixing coefficients. The choice of choosing the appropriate coefficient is, however, very important and very difficult (e.g., Burchard et al., 2005)."*

**Ute Daewel**: Even though I found the ms well readable, the English needs substantial improvement. There are several occasions with wrong grammar, typos and somewhat odd expressions. Some are listed below, but, not being a native speaker myself, I would suggest the authors to edit the language thoroughly.

*The authors: We followed this suggestion. We have checked the manuscript again for spelling errors and odd expression and applied more than a hundred corrections.*

**Ute Daewel**: specific comments: p2, l 6: "[...] a reduction of loads will have no net effect on the nutrient budget because cyanobacteria will compensate…" Can you give a citation?

*The authors*: *This is a logical consequence in models which assume a fixed Redfield-ratio. We added the reference: Molot et al. (2014). Page 2, line 5f now reads:*

*"A contrary view suggests that reduced loads will decrease primary productivity because nitrogen fixation is capped and cannot fully compensate reductions in nutrient loads (Molot et al., 2014)".*

**Ute Daewel**: p2, l 12-16: "Some of the numerous studies on cyanobacteria [...] are motivated by concerns to run into low-oxygen conditions [...]" Please consider reformulation

*The authors*: *The sentence was reformulated for clarification. Page 2, line 10ff now reads:*

*"Some of the numerous studies on cyanobacteria, which we will review in this study, are motivated by concerns to run into low-oxygen conditions triggered by global warming: warming decreases the solubility of oxygen in seawater which leads to lower oxygen concentrations."*

**Ute Daewel**: p2, l 22: comma
*The authors*: *Thanks.*

**Ute Daewel**: p2,l 30: reference from 2006: can you add a more up to date reference

*The authors*: *Agreed. We added Shimoda et al., 2016 and Taranu et al., 2012 (see below). Page 2, line 34ff now reads:*

*"Despite the importance of cyanobacteria for the Baltic Sea ecosystem, the processes involved in the bloom formation of cyanobacteria are still not comprehensively understood (Hense and Beckmann, 2006; Shimoda and Arhonditsis, 2015; Taranu et al., 2012)."*

**Ute Daewel**: p3,l 30 remove "so-called" as prognostic is a clearly defined concept A:

*The authors*: *Agreed. Page 4, line 1ff now reads:*

*"CEMBS, ECOSMO, ERGOM, SCOBI and BALTSEM are all mechanistic models as opposed to statistical models. They are, essentially, a set of partial differential equations that describe the temporal evolution of prognostic entities of relevance or interest."*

**Ute Daewel**: p4,l 7 I doubt that any modeler really does "ad-hoc" choices on parameters, please rephrase also see comment above.

*The authors*: *We rephrased the sentence. Page 4, line 12f now reads:*

*"Typically, the respective parameters and formulations are based on abductive reasoning which introduces substantial uncertainty to the realism of the model dynamics."*

**Ute Daewel**: p4, l 23ff: "[...] cyanobacteria grow more slowly [...] and can in most models only thrive when nitrogen is no longer accessible to ordinary phytoplankton [...]" The citations contain both modelling and observational studies. Does this assumption relate to models or observations?

*The authors*: *It is an assumption implicit to most models, supported by observational evidence. We changed the references to cite only modelling studies in the revised version of our manuscript to make the context clearer. Page 4, line 28-31 now reads:*

*"A basic concept of the current generation of biogeochemical models is generally the widespread paradigm that diazotrophic cyanobacteria grow more slowly than ordinary phytoplankton and can, therefore, in most models only thrive when nitrogen is no longer accessible to ordinary phytoplankton (LaRoche and Breitbarth, 2005; Hense and Beckmann, 2006; Deutsch et al., 2007)."*

**Ute Daewel**: p4,l 29-31 Can you give a reference where this was done. The phytoplankton bloom dynamics is generally determined by nutrient availability, which is the obvious reason for the nitrogen depletion in surface waters.

*The authors*: *True – we added a sentence on bottom-up control. Page 4, line 32ff & page 5, line 1-3 now reads:*

*"The phytoplankton bloom dynamics is generally determined by nutrient availability, which is the obvious reason for the nitrogen depletion in surface waters. Losses to phytoplankton abundances are set by sink terms which are designed to account for viral lysis, extracellular release, and zooplankton grazing. All models considered here resolve one functional group of zooplankton, with the exception of ECOSMO-model, which resolves two (both,*

*micro- and macro-zooplankton). As a general rule the model parameters associated to zooplankton growth (fuelled by grazing on phytoplankton) are tuned such that phytoplankton losses exceed the growth, which ultimately leads to a termination of blooms at the right time of the year."*

**Ute Daewel**: P5, l 6 remove "they"
***The authors****: Thanks.*

**Ute Daewel**: P5, l15-16 please rephrase, its not clear which model does what.
***The authors****: We changed the sentence in the revised version of the manuscript to clarify the differences between the models. Page 5, line 21-25 now reads:*

*"The respective functional forms and thresholds, however, differ between models - with the ERGOM model requiring the highest temperatures to permit growth (Fig. 1a). The SCOBI and ECOSMO-models include an additional switch which shuts down cyanobacterial growth at salinities above 10 and 11.5 PSU, respectively. Yet another level of complexity is added in SCOBI where growth necessitates oxygen concentrations above 0.1 mlO2=l , with growth gradually increasing above this oxygen threshold."*

**Ute Daewel**: P5, l20 comma
***The authors****: Thanks.*

**Ute Daewel**: P9 Section 3: In this sections there are some paragraphs that basically just list the same numbers that are given in the tables. You might want to consider shortening these paragraphs to what is new and necessary and avoiding listing the same numbers in the text and in a table.
***The authors****: Agreed. We have shortened the sentences, removed superfluous information, and listed the content in the form of bullet points. Please find our changes at:*
*3.1.1. Maximum growth. Page 10, line 11-17 now reads:*
*Reformatted into bullet points to improve readability.*

*3.1.2. Temperature dependency. Page 11, line 1-9 now reads:*
*" – Aphanizomenon flos-aquae has a somewhat wider optimal temperature range than Nodularia spumigena, spanning from 16-31°C (Bugajev et al., 2015; Carey et al., 2012; Robarts and Zohary, 1987; Paerl and Otten, 2013; Degerholm et al., 2006). Temperatures which permit some growth are considerably lower than the optimal temperatures: Cirés and Ballot (2016) report that Aphanizomenon flos-aquae is able to grow at temperatures down to 10°C. Üveges et al. (2012) measured intensive photosynthesis at even lower 2-5 °C."*
*– For Dolichospermum sp. optimal temperatures for maximal growth lie between 18 and 25°C. Growth starts at 10°C, as reported by (Hellweger et al., 2016; Konopka and Brock, 1978; Robarts and Zohary, 1987; Paerl and Otten, 2013).*
*– Nodularia spumigena prefers 20-25 °C for optimal growth (Degerholm et al., 2006). Growth start at 5 °C (Nordin et al., 1980)."*

*3.1.4. Light limitation. Page 14, line 3-15 now reads:*
*" – For Aphanizomenon flos-aquae optimal irradiance for photosynthesis are reported to be 6->33 Wm$^{-2}$ in laboratory and field experiments (Lehtimäki et al., 1997; Üveges et al., 2012). Photoinhibition for Aphanizomenon flos-aquae was reported at a light intensity of 99 Wm$^{-2}$ (Pechar et al., 1987).*
*– According to Eigemann et al. (2018) and Walsby and Booker (1980), Dolichospermum sp. prefers rather low irradiance values of 4 - 8 Wm$^{-2}$.*
*– Several scientists tested the optimal light intensity for Nodularia spumigena. They state that Nodularia spumigena grows best at the highest light level tested (23 - 66 Wm$^{-2}$). Net growth was reported over a wide range of light intensities 5-114 Wm$^{-2}$ in laboratory experiments (Eigemann et al., 2018; Jodlowska and Latala, 2019; Lehtimäki et al., 1997; Nordin et al., 1980; Roleda et al., 2008). Nordin et al. (1980) observe that very high temperatures (35°C) in combination with high light levels (114 Wm$^{-2}$) can prohibit the growth of Nodularia spumigena. Consistently, Jodlowska and Latala (2019) report reduced filament concentration and reduced photosynthesis rates at a combination of high light intensities and high temperatures (30°C). Noteworthy is, that Jodlowska and Latala (2019) did not find any photoinhibition of Nodularia spumigena until 153 Wm$^{-2}$."*

*3.1.5. Buoyancy. Page 15, line 11-19 now reads:*

*" – For Aphanizomenon flos-aquae very different floating velocities of 4-52 m d⁻¹ (Adam, 1999; Reynolds, 2006; Walsby et al., 1995) were measured. Walsby et al. (1995) also observed, that the floating velocities of Aphanizomenon flos-aquae vary depending on light conditions, with higher velocities under low light.*
*– In comparison Dolichospermum sp. is assumed to float much slower, as it does not form aggregates (Stal et al., 2003). Accordingly, Walsby et al. (1995) report respective floating velocities of 0.1-0.3 m d⁻¹ for Dolichospermum sp.. Note in this context, that Dolichospermum sp. occurs mainly at depth of 5-10 m.*
*– In the Baltic Sea, Nodularia spumigena tends to float to the sea surface and there is no evidence for large changes in floating velocities under varying light conditions (Adam, 1999). Recorded floating velocities range from 35-46 m d⁻¹ for Nodularia spumigena colonies (Adam, 1999; Walsby et al., 1995)."*

*3.3.1. Salinity constrains. Page 23, line 14-28 now reads:*

*" – Aphanizomenon flos-aquae is known as a freshwater species (Laamanen et al., 2002). Accordingly, Rakko and Seppäälä (2014) and Laamanen et al. (2002) measured rather low salinities of 0-5 PSU for optimal growth. Rakko and Seppäälä (2014) describe this species as rather coastal, preferring less saline conditions. In line, Lehtimäki et al. (1997) state that Aphanizomenon flos-aquae is not able to tolerate salinities higher than 10 PSU. Consistently, its abundance decreases from the northern to the southern part of the Baltic proper.*
*– The taxa Dolichospermum originate from freshwater, with some strains adapted to brackish water (Brutemark et al., 2015). In agreement, in the Baltic Sea the specie Dolichospermum flos-aquae shows similar growth rates between 0-10 PSU and a strong decrease in growth rates at higher salinities (Moisander et al., 2002).*
*– For the species Dolichospermum aphanizomenoides, Moisander et al. (2002) report a wide range of salinities, from 0-10 PSU, which are related to similar growth rates. Dolichospermum taxa originate from freshwater, with some strains that are adapted to brackish water (Brutemark et al., 2015). In the Baltic Sea Dolichospermum spp. can grow in salinities between 0-10 PSU, but their growth rates decrease strongly above 10 PSU (Moisander et al., 2002).*
*– For different strains of Nodularia spumigena a wide range of tolerable salinities were reported: 0-20 PSU (Moisander et al., 2002; Lehtimäki et al., 1997), while Rakko and Seppäälä (2014) and Nordin et al. (1980) narrow the optimal salinity range down to 5-10 PSU."*

*3.3.2 Cyanobacteria akinetes. Page 24, line 17-32 now reads:*
*Reformatted into bullet points*

**Ute Daewel**: P10 l23-24 Grammar, please rephrase the two sentences. (& comma)
***The authors***: *We rephrased and shortened the sentences. Page 11, line 1-5- now reads:*
*" – Aphanizomenon flos-aquae has a somewhat wider optimal temperature range than Nodularia spumigena, spanning from 16-31°C (Bugajev et al., 2015; Carey et al., 2012; Robarts and Zohary, 1987; Paerl and Otten, 2013; Degerholm et al., 2006). Temperatures which permit some growth are considerably lower than the optimal temperatures: Cirés and Ballot (2016) report that Aphanizomenon flos-aquae is able to grow at temperatures down to 10°C. Üveges et al. (2012) measured intensive photosynthesis at even lower 2-5 °C."*

**Ute Daewel**: P11,L8 "Similar to this,..."
***The authors***: *Thanks*

**Ute Daewel**: P15,L28-29 Please explain this sentence.
***The authors***: *We rephrased the sentence to improve the explanation of the link between PCD and biogeochemical cycling. Page 16, line 31ff now reads:*
*"Also, cell death by PCD may facilitate biogeochemical cycling, through the regular death of cells by PCD and the resulting release of organic and inorganic matter, including the redistribution of fixed nitrogen (Berman-Frank et al., 2004)."*

**Ute Daewel**: P18,L27 Reference?

*The authors*: *see reference Eglite et al., 2019/ Wasmund et al., 2019 added to page 20, line 6ff now reads:*

*"Among the most important and abundant grazers in the Baltic Sea are copepods. Typical species of copepods in the Baltic Sea are e.g. Acartia logiremis, Temora longicornis or Centropages hamatus (Eglite et al., 2019; Wasmund et al., 2019)".*

**Ute Daewel**: P19,L5 "...,they are not able to..."

*The authors*: *Thanks. Page 20, line 18f now reads:*

*"Overall, while copepods play an important grazing role in the ecosystem, they are not able to control cyanobacteria growth in the Baltic Sea (Sommer et al., 2006)."*

**Ute Daewel**: P22, l14-15 revise sentence

*The authors*: *Thanks. Please see: Page 23, line 26ff now reads:*

*"For different strains of Nodularia spumigena a wide range of tolerable salinities were reported: 0-20 PSU (Moisander et al., 2002; Lehtimäki et al., 1997), while Rakko and Seppäälä (2014) and Nordin et al. (1980) narrow the optimal salinity range down to 5-10 PSU."*

**Ute Daewel**: p23, l27: "Sect. 4.2 debates the impact of the oceanic processes to the Baltic Sea, respectively" In respect to what?

*The authors*: *We deleted ",respectively" behind the word "Baltic Sea in the following sentence. Page 25, line 10f now reads:*

*"Sect. 4.2 debates the impact of the oceanic processes to the Baltic Sea. Some considerations about model assessment metrics are in sect. 4.3."*

**Ute Daewel**: p24, l9: "Another potentially problematic assumption [...] is the fixed Redfield-ratio[...]" Why and how is this problematic? Enhance

*The authors*: *e.g., page 13, L 1-10 refers to the cyanobacteria's storage capacity of DIP which is very difficult to account for when assuming a fixed Redfield-ratio. Not considering this in a model might shrink the ecological niche of cyanobacteria. Further, there is observational evidence that diazotrophs do not obey Redfield (e.g., Larsson et al, 2001). We rephrased the sentence and made the context clear in the revised version of the manuscript:*

*Chapter 4.1 Biogeochemical processes. Page 25, line 25-28 now reads:*

*"Another potentially problematic assumption which might introduce substantial uncertainty in model-based projections of nutrient load scenarios is the fixed Redfield-ratio engrained in most models, and which does not account for the storage capacity of DIP by cyanobacteria. This might spuriously shrink their ecological niche."*

**Reviewer #2, Justus van Beusekom:**

Dear Justus van Beusekom,
thank you very much for your time and effort. Based on your (and the other reviewer's help) we rewrote so much that we invented the following system in our point-by-point responses: New, revised text is marked in font colour green. Blue denotes the original text into which we embedded our (green) changes.

Kind regards,
the authors

**Justus Van Beusekom**: In this paper, the authors discuss Cyanobacteria blooms in the Baltic. These blooms may cause environmental problems including toxic events or large biomass productions leading to coastal anoxia. A correct modelling of such blooms is important to check measures to combat eutrophication. The authors aim to bring together both a modelers view and a biologists view. They compare 5 models that are used for political decision making regarding the approach to model cyano-blooms. Against this background, the factors that determine the cyanobacteria blooms in the real world are discussed. Based on the comparison of models and field/lab studies, the authors conclude that modelers tend to keep models simple (with good reasons), whereas in the real world cyano-blooms are complex with several species each with their specific requirements responding differently to changes in drivers (like nutrients, light, temperature, grazing).
The paper is well written, but the discussion and outlook needs more substance. It does not come as a surprise that modelers tend to (have to) keep models simple and that biologists have an eye on complexity. Given this, I expect a more in-depth discussion on the next steps to be taken to overcome this schism. For instance, given the focus on those models that are used for policy purposes, the limits of using these models should be discussed and suggestions should be made how this dilemma can be solved. Could for instance the combination of "simple" biogeochemical models and conceptual models be a solution? And are examples available?

*The authors*: *We find your suggestions very constructive and changed the discussion and outlook accordingly. Specifically, we included an additional chapter: '4.4. Future Perspectives' to assess the possible impact of the formulation of cyanobacteria on future projections and make (now) suggestions on how to proceed to solve the dilemma in our revised manuscript. Further, we reformulated the "Conclusions", which now explains the difficult situation of modelers and suggests measures to overcome these.*

*Specifically, we added:*

*4.4 Future Perspectives. Page 27, line 22-32 & page 28, line 1-34 now reads:*

[revised manuscript text omitted]

Minor comments:
***Justus Van Beusekom****: P1L12, delete commas around ": : :.severe"*

*The authors*: Thanks. Page 1, line 12 now reads:
*"One particularly severe problem is eutrophication."*

**Justus Van Beusekom**: P2L4. This sentence is not very clear but very important. I would add a sentence, that explains the main question: How will future, management-induced changes in nutrient loads affect the blooms and how does this interact with expected warming?
*The authors*: We addressed this question in chapter '4.4. Future Perspectives' (see answer above; Page 27, line 22-32 & page 28, line 1-34)

**Justus Van Beusekom**: P2L11 Phosphorus (not ..ous) P1L19: Sedimentary processes: this needs some explanation. Especially feedbacks between cyanos and P-release should at least be mentioned.
*The authors*: Thank you for your suggestion, we included a paragraph about sediment processes and P-release in our revised manuscript. Page 2, line 18-22 now reads:
*"One problem that makes consequences of this chain of events so unpredictable and a precise quantitative process understanding so desirable is the existence of a positive feedback loop where low-oxygen conditions may drive P-release from the sediments. This excess P (which comes without the Redfield N equivalent to the system) may fuel additional cyanobacteria blooms (Conley et al., 2002; Savchuk, 2018; Stigebrandt et al., 2014; Vahtera et al., 2007b) thereby closing the positive feedback loop."*

**Justus Van Beusekom**: P3L4. Alternatives: so, in the end I expect suggestions on how superior alternatives can be developed.
*The authors*: Following your suggestions we added the chapter '4.4. Future Perspectives' (see answer above. Page 27, line 22-32 & page 28, line 1-34)

**Justus Van Beusekom**: P4L6: prognostic variables: add s
*The authors*: Thanks.

**Justus Van Beusekom**: P5L31: Typo in .19_C? (dot 19)
*The authors*: Thanks.

**Justus Van Beusekom**: P6L32 PAR instead of RAR
*The authors*: Thanks.

**Justus Van Beusekom**: P11L30 POsubscript4, superscript 3 (you swapped both)
*The authors*: Thanks.

**Justus Van Beusekom**: P13, header: You discuss Pmax, but no mention is made of alpha. I suggest to add at least some words to this.
*The authors*: We rewrote the paragraph and have included some explanations about alpha in our revised manuscript. Page 13, line 12-21 & page 12, line 27-34 now reads:
*"Cyanobacterial and algal photosynthesis rates are significantly influenced i.a. by the combination of light intensity and temperature (Butterwick et al., 2005). With increasing light intensity photosynthesis rate will increase until the saturation level is achieved (Ik) and the maximal photosynthesis rate (Pmax) is reached. The initial slope of this photosynthesis-irradiance curve (alpha) describes the performance of both light-harvesting and photosynthetic conversion efficiency. Alpha is species specific and rather temperature-independent over a wide range. Alpha is, however, a strong function of the highly variable carbon specific chlorophyll a content of cells owing to the central role of chlorophyll in photosynthesis.*
*Pmax, on the other hand, will be influenced i.a. by temperature. Below the temperature optima, P max increases non-linearly roughly doubling with each 10 _C rise in temperature until a threshold temperature (Reynolds, 2006). Beyond the threshold excessive temperatures in combination with prolonged exposure to high light intensities may cause photo-inhibition and induce harmful effects in algae (Ibelings, 1996)."*
*............."The species Nodularia spumigena and Aphanizomenon sp. seem to be well acclimated to relatively high PAR levels, especially at high temperatures ( Sliwinska-Wilczewska et al., 2019). Their cell-specific Pmax values were the highest in Nodularia spumigena and Aphanizomenon sp. strains grown under the lowest light*

*intensity. Both species changed their Chl a -specific alpha depending on environmental conditions. Maximum alpha values for Nodularia spumigena  and Aphanizomenon sp. were found at low light, low temperature and low salinity  (10 μmol ∗ photons ∗ m$^{-2}$; 15°C  and 8 PSU) ( Sliwinska-Wilczewska et al., 2019). Similar to this DeNobel et al. (1998) found that for Aphanizomenon flos-aquae  and Anabaena sp. their alpha increased with decreasing irradiance but was always higher for Aphanizomenon flos-aquae  then for Anabaena sp. . This was due to a higher chlorophyll a content in cells of Aphanizomenon flos-aquae than in Anabaena sp. .”*

**Justus Van Beusekom**: P13L13 for Nodularia: space between the words
***The authors***: *Thanks.*

**Justus Van Beusekom**: P14L11: Should this be Table 10?
***The authors***: *It's great that you noticed this. We corrected the reference to the table. Page 15, line 9f now reads:*
*“Listed below are the floating velocities for the considered cyanobacteria species of the Baltic Sea (Table 10 specifies the respective individual experiments).”*

**Justus Van Beusekom**: P15L18: Claessen et al describe (no s)
***The authors***: *Thanks.*

**Justus Van Beusekom**: P18L9 Inline -> In line ? (see also P21L33)
***The authors***: *Thank you for this note. We corrected all wrong "in line" in the revised version.*

**Justus Van Beusekom**: P18 L28: hamatusarexs ->hamatus?
***The authors***: *Thank you for this note. You are correct with this species name. We changed the species name. Page 20, line 7ff now reads:*
*“Typical species of copepods in the Baltic Sea are e.g. Acartia logiremis , Temora longicornis  or Centropages hamatus  (Eglite et al., 2019; Wasmund et al., 2019).”*

**Justus Van Beusekom**: P19L24: exceed -> excert?
**The authors**: Thanks.

**Justus Van Beusekom**: P21L31: specie ->species
***The authors***: *Thanks.*

**Justus Van Beusekom**: P21L32: add space after (2002)
***The authors***: *Thanks.*

**Justus Van Beusekom**: P23L27: processes on the Baltic. (delete respectively
***The authors***: *Thanks, we deleted the respectively.*

**Justus Van Beusekom**: P25L33: politic->policy?
***The authors***: *That is right, thank you.*

**Justus Van Beusekom**: P25L14: "A comparison of". Do you refer to the present study?
***The authors***: *Yes, we refer to models and field experiments studied in this manuscript. We changed the whole conclusion, such that it is easier to understand. Page 29, line 12-15 now reads:*
*Further spread in the underlying assumptions is introduced by the, necessarily, reductionist approach of modelers, which often contrasts with the more complex findings of ecologist – a problem especially prominent because the most dominant cyanobacteria species of the Baltic Sea are diverse and feature very differing traits (cf., 4.4 Future  Perspectives). To this end, the overarching question is how much complexity is needed for reliable projections.*

**Justus Van Beusekom**: P25L17ff: an unclear sentence. Maybe delete at the end "of which"?
***The authors***: *That is right, thank you. We changed and shortened the whole chapter and deleted the sentence (See last answer for the conclusion. Page 29, line 1-22).*

**Comments #3, Oleg Savchuk:**

Dear Oleg Savchuk,
thank you very much for your time and effort. Based on your (and the other reviewer's help) we rewrote so much that we invented the following system in our point-by-point responses: New, revised text is marked in font colour green. Blue denotes the original text into which we embedded our (green) changes.

Kind regards,
the authors

**Oleg Savchuk**: In my mind, this manuscript provides a very interesting and thorough coverage of an important topic.
*The authors*: *We thank Oleg Savchuk for this encouraging statement!*

**Oleg Savchuk**: Although I cannot quite agree with some of the author's statements related to modelling approach, the following notes of mine are by no means a formal review of the manuscript, but merely a correction of some confusion and/or misunder-standing.1) In BALTSEM model, all the variables and biogeochemical fluxes are expressed in the weight, not molar units. As indicated in Table B2 (Savchuk, 2002), the phosphate half-saturation constants for the cyanobacteria growth and nitrogen fixation rates are 1.5 and 9.0 mg P m$^{-3}$, that is about 0.05 and 0.3 mmol P m$^{-3}$, respectively. Please, in your Table 03; at p. 11, lines 5-7; and elsewhere if necessary.
*The authors*: *Thank you very much for pointing this out! We adjusted Tab.3 and added a remark to the caption that the provided values for the BALTSEM-model were unit converted (compared to the original study). Additionally, the overall range of half-saturation constants used by the models changed slightly and we adjusted this in Sect. 3.1.3 Nutrient demands. Page 11, line 19 now reads:*
*"The respective values for the half saturation constants envelope a large range from 0.05 - 0.5 mmolP/m3 (c.f., Table 3)."*

**Oleg Savchuk**:  2) In BALTSEM model, both the mortality rates and sinking velocities of phytoplankton groups depend not only on the water temperature but are also inversely related to the Liebig minimum function as a measure of unfavorable environmental conditions; for cyanobacteria, accounting also for contribution of nitrogen fixation into their total primary production determined by both DIN uptake and nitrogen fixation. In result, neither mortality rate nor sinking velocity have a fixed value, contrary to what is now stated at p. 15, lines 4-6.
*The authors: We apologize and corrected this in the captions of Table 3 (page 54) and Table 4 (page 55) and in Sect.3.2., page 16, line 5ff now reads:*

*Tab 3: "* Note that, as a peculiarity, the BALTSEM models uses different sinking rates for diatoms in spring and autumn and also the sinking speed depends on temperature, environmental conditions *and morphology."*

*Tab 4: "* The phytoplankton mortality rates in BALTSEM *depend on temperature* and the environmental *conditions."*

*Sect.3.2.*
*"For all non-grazing related losses, the models generally assume a fixed loss rate which depends either linearly or quadratic on abundance. An exception is the BALTSEM model, where the mortality of phytoplankton depends on water temperature and is also inversely related to the Liebig minimum function (as a measure of unfavourable environmental conditions (c.f., Sect.2.3)."*

---

## Author Response (AR3)

Kiel, 12th of Feb., 2021
Dr. Britta Munkes Dr. Ulrike Löptien, Dr. Heiner Dietze
Helmholtz Centre for Ocean Research Geomar
Düsterbrooker Weg 20
D - 24105 Kiel
bmunkes@geomar.de; uloeptien@geomar.de; hdietze@geomar.de

Dear Perran Cook,

thank you for your positive comments to our last submitted manuscript. Now we have included the technical corrections (please see next page) and submitted the manuscript once more.
We hope, that we have corrected all typos to your satisfaction.

Yours Sincerely,
The authors

Corrections of the manuscript:

**Perran Cook**:
Pg2 Line 10 This sentence is unclear. …are motivated by concerns about low-oxygen condition….
***The authors***: *We rephrased the first sentence, which were unclear. Page 2, line 10f now reads:*
Some of the reviewed studies on cyanobacteria assume that global warming will exacerbate the existing oxygen deficiency in the Baltic Sea. Warming decreases the solubility of oxygen in seawater which leads to lower oxygen concentrations. Further, warming conditions may favour cyanobacteria because they are better adapted to oligotrophy and they benefit from the increased light levels that come along with increased stratification in response to increased air-sea heat fluxes (Carey et al., 2012; Paerl and Huisman, 2009; Andersson et al., 2015b). Increased nitrogen fixation helps to overcome oligotrophy and increases primary production and subsequent export of organic matter to depth.

**Perran Cook**:
Pg 18 line 14 reword to 'there is little evidence'
***The authors***: *We replaced the wording: 'there are only few evidences' in 'there is little evidence. Page 18, line 14f now reads:*
The importance of viruses in controlling cyanobacteria abundances is, however, still poorly understood. There is little evidence for the impact of viruses on cyanobacteria.

**Perran Cook**:
3.3.1 salinity constraints
***The authors***: *We deleted the duplicate paragraph on Dolichospermum sp. Page 23, line 28ff and an unnecessary sentence about Nodularia sp. on  Page 23, line 20f. The paragraph now reads:*

Salinity constrains
The Baltic Sea features a wide range of salinities, ranging from 15-25 PSU in the north western part of the Baltic to 2-3 PSU in the Bothnian Bay. The Baltic Proper, situated in the centre, is characterised by intermediate values around 6-8 PSU. The large spatial variance in salinity can induce large local salinity variations over time when ocean currents mix water masses from different origins. This can decrease the local growth- and photosynthesis rates of algae and cyanobacteria, once specific salinity thresholds are over- or undercut and physiological stress sets in. By this means-salinity has the potential to control the occurrence of cyanobacteria species. Salinity thresholds are set in SCOBI and ECOSMO where growth is not permitted above 10 and 11.5 PSU, respectively. The other models do not include salinity constrains on simulated cyanobacteria.

Field observations show that in most parts of the Baltic Sea large cyanobacteria blooms occur during summer, except in the relatively saline waters in the Kattegat and the Belt Sea. Thus, e.g., Rakko and Seppäälä (2014) conclude, in line with the SCOBI and ECOSMO  models, that high salinities seem to restrict the growth of Baltic Sea cyanobacteria and estimate a threshold around 10 PSU. Low salinities, in contrast, do not seem to restrict the growth of cyanobacteria in the Baltic Sea in general and several studies report blooms at very low values: Wasmund (1997) relates Baltic Sea blooms to salinities between 3.8-11.5 PSU and Kononen et al. (1996) report no significant correlation of bloom-forming cyanobacteria in the Gulf of Finland with salinity (i.e., salinities between 3-6  PSU).

The differences between species are substantial. Brutemark et al. (2015) even state, that salinity might be one of the main factors that explains the distribution of species in the Baltic Sea. These statements are supported by laboratory experiments which we list in the following for the most relevant species.

> – *Aphanizomenon flos-aquae* is known as a freshwater species (Laamanen et al., 2002). Accordingly, Rakko and Seppäälä (2014) and Laamanen et al. (2002) measured rather low salinities of 0-5 PSU for optimal growth. Rakko and Seppäälä (2014) describe this species as rather coastal, preferring less saline conditions. In line, Lehtimäki et al. (1997) state that *Aphanizomenon flos-aquae* is not able to tolerate salinities higher than 10 PSU. Consistently, its abundance decreases from the northern to the southern part of the Baltic proper.

– The taxa Dolichospermum originate from freshwater, with some strains adapted to brackish water (Brutemark et al., 2015). In agreement, in the Baltic Sea the specie Dolichospermum flos-aquae shows similar growth rates between 0-10 PSU and a strong decrease in growth rates at higher salinities (Moisander et al., 2002).

– For different strains of *Nodularia spumigena* a wide range of tolerable salinities were reported: 0-20 PSU (Moisander et al., 2002; Lehtimäki et al., 1997), while Rakko and Seppäälä (2014) and Nordin et al. (1980) narrow the optimal salinity range down to 5-10 PSU.

To sum up, in most areas of the Baltic Sea, salinity is no restriction for growth of cyanobacteria (an exception are the Danish Straits). Interestingly, however, salinities apparently affect the toxicity of cyanobacteria blooms: Mazur-Marzec et al. (2005) report an increase of Nodularin production for *Nodularia spumigena* with increasing salinity concentrations. In line, Brutemark et al. (2015) found the highest intracellular toxin concentration at the highest tested salinity concentrations (6 PSU) for *Dolichospermum spp.*.